# The plasma lipidome of the Quaker parrot (*Myiopsitta monachus*)

Hugues Beaufrère[1]*, Sara M. Gardhouse[2¤], R. Darren Wood[3], Ken D. Stark[4]

1 Department of Clinical Studies, Ontario Veterinary College, University of Guelph, Guelph, Ontario, Canada,
2 Health Sciences Centre, Ontario Veterinary College, University of Guelph, Guelph, Ontario, Canada,
3 Department of Pathobiology, Ontario Veterinary College, University of Guelph, Guelph, Ontario, Canada,
4 Department of Kinesiology, University of Waterloo, Waterloo, Ontario, Canada

¤ Current address: Department of Clinical Sciences, College of Veterinary Medicine, Kansas State University, Manhattan, Kansas, United States of America
* beaufrer@uoguelph.ca

**Data Availability Statement:** Data are available in the Scholars Portal Dataverse (doi: 10.5683/SP2/XUW31U).

**Funding:** HB: D19ZO-301, Morris Animal Foundation, https://www.morrisanimalfoundation.org

## Abstract

Dyslipidemias and lipid-accumulation disorders are common in captive parrots, in particular in Quaker parrots. Currently available diagnostic tests only measure a fraction of blood lipids and have overall problematic cross-species applicability. Comprehensively analyzing lipids in the plasma of parrots is the first step to better understand their lipid metabolism in health and disease, as well as to explore new lipid biomarkers. The plasma lipidome of 12 Quaker parrots was investigated using UHPLC-MS/MS with both targeted and untargeted methods. Targeted methods on 6 replicates measured 432 lipids comprised of sterol, cholesterol ester, bile acid, fatty acid, acylcarnitine, glycerolipid, glycerophospholipid, and sphingolipid panels. For untargeted lipidomics, precursor ion mass-to-charge ratios were matched to corresponding lipids using the LIPIDMAPS structure database and LipidBlast at the sum composition or acyl species level of information. Sterol lipids and glycerophospholipids constituted the majority of plasma lipids on a molar basis. The most common lipids detected with the targeted methods included free cholesterol, CE(18:2), CE(20:4) for sterol lipids; PC (36:2), PC(34:2), PC(34:1) for glycerophospholipids; TG(52:3), TG(54:4), TG(54:5), TG (52:2) for glycerolipids; SM(d18:1/16:0) for sphingolipids; and palmitic acid for fatty acyls. Over a thousand different lipid species were detected by untargeted lipidomics. Sex differences in the plasma lipidome were observed using heatmaps, principal component analysis, and discriminant analysis. This report presents the first comprehensive database of plasma lipid species in psittacine birds and paves the way for further research into blood lipid diagnostics and the impact of diet, diseases, and drugs on the parrot plasma lipidome.

## Introduction

Dyslipidemias and lipid-accumulation disorders, such as atherosclerosis, hepatic lipidosis, fatty tumors and obesity, are extremely common in captive Psittaciformes [1–4]. The prevalence of severe atherosclerotic lesions in the general parrot population is approximatively 7%, but can be as high as 50% in older parrots [3]. Hepatic lipidosis is also prevalent and one of the

This publication has not been reviewed or endorsed by the Morris Animal Foundation and the views expressed do not necessarily reflect the views of the Foundation, its officers, directors, affiliates or agents. HB: 054209, OVC Pet Trust, https://ovc.uoguelph.ca/pettrust/help-pets-we-love-live-longer-healthier-lives The funders had no role in study design, data collection and analysis, decision to publish, or preparation of the manuscript.

**Competing interests:** The authors have declared that no competing interests exist.

major liver diseases in parrots with an estimated overall prevalence of 6%, but susceptible species such as Quaker parrots have an estimated prevalence of 20% [2]. The prevalence of female parrot reproductive disorders associated with upregulated vitellogenesis (hepatic lipid synthesis and lipid transport to eggs) is unknown, but is suspected to be enormous based on clinical experience. Taken together, lipid-related disorders are likely one of the most common causes of non-infectious diseases in captive parrots. However, despite the frequency of these diseases, there is a vast gap of knowledge in regards to the pathophysiology of most of these diseases as well as diagnostic tests, biomarkers, treatment, and therapeutic targets. As dyslipidemic changes are frequently concurrent and comorbid to lipid-accumulation disorders as well as important risk factors, blood lipid analysis can be used for the diagnosis, screening, and monitoring of a variety of diseases associated with lipid dysmetabolism [1, 5–7].

Lipids are the main biomolecular constituents of plasma [8] and are transported in the form of macromolecular aggregates of mixed lipid species and proteins (lipoproteins). Traditionally, dyslipidemias in mammals have been understood as primarily associated with elevated total cholesterol, triglycerides and changes in cholesterol lipoprotein fractions [9, 10]. While psittacine lipoproteins have been measured in plasma using reference methods [11–13], routine laboratory methods have not been validated in parrots and are likely to perform poorly due to the marked differences between avian and mammalian lipoprotein structure and metabolism [14, 15]. For these reasons, lipoprotein testing has not been widely applied to parrots with dyslipidemia or lipid-related disorders and the lipoprotein profiles of psittacine spontaneously-occurring dyslipidemia have not been characterized.

In order to elucidate the pathophysiology of the many common lipid-related diseases in parrots as well as to provide new tools for biomarkers discovery, a more comprehensive analysis of their lipid metabolism in health and disease is required. Comprehensive lipid analysis by mass spectrometry is known as lipidomics. This approach is revolutionizing the way lipid metabolic disorders are investigated as a results of the vast amount of data generated when blood lipids are analyzed using lipidomics [16–20]. Further, in the context of dyslipidemia, specific lipid species of complex lipids may be better targets or biomarkers than crude measurements of a single lipid molecule such as cholesterol (the cholesterol test measures both free cholesterol and the cholesterol moiety of a variety of cholesteryl esters) and triglycerides (the triglyceride test only measures the glycerol backbone of the molecule). Lipidomics allows for the measurement of each lipid in their native biological form. Therefore, thousands of species of fatty acyls, triglycerides, phospholipids, sphingolipids, and cholesteryl esters with a variety of saturated and unsaturated fatty acid chains can be measured in plasma. In humans, lipidomics has been used to study the lipidome [8, 21], dyslipidemia [22, 23], various clinical lipid-accumulation disorders [17, 19, 20], and statin pharmacology [24, 25]. However, this powerful analytical technique has not been applied to psittacine birds or used in avian health research as far as the authors know. Plasma lipidomic profiling in psittacine birds may not only provide a clearer picture of ongoing lipid abnormalities, but also lead to innovation in lipid biomarkers and therapeutic targets beyond cholesterol for a variety of lipid-related diseases.

The first step in applying clinical lipidomics in psittacine medicine is to report the plasma lipidome as blood is the most accessible tissue and plasma biomarkers are the most practical to develop. Knowing the normal lipidome may also allow to detect specific lipidomic signatures of lipid-related diseases in birds. The Quaker parrot (*Myiopsitta monachus*) has been used as an experimental model of lipid disorders in Psittaciformes for dyslipidemia and atherosclerosis [11, 12, 26, 27]. Quaker parrots are also extremely prone to spontaneously-occurring dyslipidemia and lipid-accumulation disorders, more than other psittacine species [2, 4]. It is therefore logical to first use this species to report the psittacine plasma lipidome. The objective of this observational study was to comprehensively report the plasma lipidome of young male and

female healthy Quaker parrots using a variety of quantitative (targeted lipidomics) and semi-quantitative (untargeted lipidomics) methods.

## Materials and methods

### Animals and sample collection

Twelve approximately 1-year-old Quaker parrots (*Myiopsitta monachus*) were used for this study. The parrots were captive-bred and hand raised at the Hagen Avicultural Research Institute (QC, Canada). The parrots included 6 males and 6 females; sex was confirmed by DNA testing on blood. The parrots were housed together at the University of Guelph–Central Animal Facility in a large stainless-steel aviary with food and water provided *ad libitum*, and fed a pelletized diet (Tropican 2mm pellet, Hagen Inc., Baie d'Urfee, QC, Canada). They were considered healthy and free of dyslipidemia based on a recent physical examination, CBC, plasma biochemistry, lipoprotein panel, and avian chlamydiosis PCR testing. Animal utilization protocols (AUP) were approved for this research by the University of Guelph—Animal Care Committee (AUP#3875 and AUP#4035).

The parrots were fasted overnight prior to sample collection. Blood was collected and stored according to guidelines for plasma lipidomics [28]. To minimize stress and exertion, birds were captured in the dark and blood was collected within 2–5 minutes following capture. A 1 mL blood sample was collected from each parrot from the right jugular vein under manual restraint using a 3mL syringe connected to a 26g needle. Blood was transferred to a heparinized tube without a serum separator (BD Microtainer, Becton and Dickinson, Mississauga, ON, Canada). Tubes were inverted a minimum of 5 times and placed on ice. Blood was centrifuged for 10 minutes at 1500g and approximately 0.5 mL plasma harvested and aliquoted in cryovials. The plasma was stored at -80C until shipping on dry ice to the various analytical laboratories.

All samples were analyzed by The Metabolomics Innovation Centre (co-located at the University of Alberta and University of Victoria, Canada). Six samples (3 females, 3 males) were submitted to the University of Victoria Genome BC Proteomics Centre for untargeted lipidomics and targeted panels for bile acids, sterols, non-esterified fatty acids, acyl carnitines, and sphingolipids. Another six samples (3 females, 3 males) were submitted to the University of Alberta for the targeted panels using a metabolomics kit for glycerolipids, glycerophosphocholines and cholesteryl esters at the University of Alberta.

### Nutritional analysis

A 100g sample of the pelleted parrot diet was submitted to an independent laboratory (SGS Canada Inc. Agriculture and Food, Mississauga, ON, Canada) for nutritional analysis. Total fat was analyzed by an acid hydrolysis method (SGS-Canada, test QAM-105) and fatty acid composition was obtained using gas chromatographic methods [Association of Official Analytical Collaboration (AOAC) International 991.39, AOAC 963.2].

### Targeted lipidomics

All lipid species were analyzed at the brutto (sum composition) or medio (fatty acyl chain) level of identification [29].

For analysis of bile acids, sterols, steroids, fatty acids, carnitines, and sphingolipids, an Agilent 1290 UHPLC system coupled to an Agilent 6495 QQQ (Agilent, Santa Clara, CA, USA) or a Sciex 4000 QTRAP (Sciex, Framingham, MA, USA) mass spectrometer equipped with an electrospray ion (ESI) source was used. The MS instruments were operated in multiple-reaction

monitoring (MRM) with negative-ion (-) detection for analysis of fatty acids and bile acids, and with positive-ion (+) detection for analysis of carnitines, sphingolipids, sterols and steroids.

**Analysis of bile acids.** Bile acids was quantitated by UPLC-MRM/MS on an Agilent 1290 UHPLC system coupled to an Agilent 6495 QQQ mass spectrometer (Agilent, Santa Clara, CA, USA), according to a previously published procedure by the University of Victoria Genome BC Proteomics Centre [30, 31]. A mixed standard solution containing reference substances of 62 bile acids, which were detailed in the same studies [30, 31], was prepared in 50% methanol at 10 nmol/mL for each compound and was used as standard solution S1. This solution was further diluted step by step at a dilution ratio of 1 to 4 (v/v) to have standard solutions of S2 to S10. Fifty μL of S1 to S10 was mixed with 50 μL of a solution containing 14 D-labeled bile acids as internal standard. Twenty μL of each solution was injected to run UPLC-(-)ESI-MRM/MS. Linear-regression calibration curves were constructed using analyte-to-internal standard peak area ratios (As/Ai) versus molar concentrations (nmol/mL) of each bile acid. For the bile acids without isotope-labeled analogues as internal standard, glycodeoxycholic acid-D4 was used as a common internal standard.

For sample preparation, 50 μL of plasma was mixed with 50 μL of the internal standard solution and 400 μL of methanol in an Eppendorf tube. After vortex mixing for 15 s and sonication for 2 min in an ice-water bath, the tube was centrifuged at 15,000 rpm for 15 min in an Eppendorf 5420R centrifuge to pellet proteins. The supernatant was taken out and dried in a nitrogen evaporator under a gentle nitrogen gas flow. The residue was dissolved in 100 μL of 50% methanol. After centrifugal clarification, 20 μL was injected for detection and quantitation of bile acids by UPLC-MRM/MS.

Concentrations of bile acids in each sample were calculated from the standard curves of the individual analytes.

**Analysis of sterols including cholesterol.** A mixed standard solution containing reference substances of 10 sterols (lanosterol, zymosterol, 7-dehydrodesmosterol, desmosterol, dihydrolanosterol, zymostenol, lathosterol, 7-dehydrocholesterol, dihydrolathosterol and cholesterol) which were obtained from Sigma-Aldrich (Oakville, ON, Canada) or Steraloids Inc. (Newport, RI, USA) was prepared in acetonitrile at 10 nmol/mL for each compound, and was used as standard solution S1. This solution was further diluted step by step at a dilution ratio of 1 to 4 (v/v) to have standard solutions of S1 to S10.

Ten μL of plasma was mixed with 90 μL of methanol. After vortex mixing for 15 s and sonication for 2 min in an ice-water bath, the tube was centrifuged at 15,000 rpm for 15 min in an Eppendorf 5420R centrifuge. The supernatant was taken out and dried in a nitrogen evaporator under a gentle nitrogen gas flow. 50 μL of acetonitrile was added to resuspend the residue.

Each sample solution or 50 μL of each standard solution, was mixed with 20 μL of an internal standard solution containing $^{13}C_3$-cholesterol. The mixture was subjected to chemical derivatization with 20 mM of dansyl chloride in the presence of 100 mM 4-(dimethylamino)-pyridine (DMAP) as a catalyst at 50 ˚C for 60 min, according to a previously published procedure with necessary modifications [32]. After derivatization, 20 μL of each resultant solution was injected to run UPLC-(+)ESI-MRM/MS on a C18 UPLC column (2.1 x 50 mm, 1.7 μm) with 0.1% formic acid and isopropanol/acetonitrile (1:2) as the mobile phase for binary-solvent gradient elution, at a flow of 0.4 mL/min and 60 ˚C.

Linear-regression calibration curves were constructed using analyte-to-internal standard peak area ratios (As/Ai) versus molar concentrations (nmol/mL) of the standard solutions for each sterol with 13C3-cholesterol as a common internal standard. Concentrations of sterols detected in each sample were calculated from the standard curves of the individual analytes.

**Analysis of selected steroids.** A mixed standard solution containing reference substances of cortisol, aldosterone and androstenedione (Steraloids Inc., Newport, RI) was prepared in

methanol at 10 nmol/mL for each compound. This solution was further diluted step by step at a dilution ratio of 1 to 4 (v/v) with the same solvent to have standard solutions of S1 to S9.

Fifty μL of plasma was mixed with 425 μL of 125 mM sodium acetate buffer (pH = 5.5) and 25 μL of an internal standard solution containing cortisol-D4 and 6β-OH cortisol-D4. After vortex-mixing, 1 mL of ethyl acetate was added and the mixture was vortex-mixed for 30 s followed by centrifugal clarification. The clear supernatant was taken out and dried in a nitrogen evaporator. The residue was reconstituted in 50 μL of methanol. Ten μL was injected to run UPLC-(+)ESI-MRM/MS on a C18 UPLC column (2.1 x 100 mm, 1.7 μm) with 0.1% formic acid and acetonitrile as the mobile phase for binary-solvent gradient elution, at 0.3 mL/min and 50 ˚C.

Linear-regression calibration curves were constructed using analyte-to-internal standard peak area ratios (As/Ai) versus molar concentrations (nmol/mL) of the standard solutions. Concentrations of steroids detected in each sample were calculated from the standard curves of the individual analytes, with internal calibration.

**Analysis of fatty acids.** Quantitation of fatty acids was performed using 3-nitrophenylhydrazine (3-NPH) derivatization–UPLC-MRM/MS, adapted from a published procedure from Han et al. [33]

A mixed standard solution containing reference substances of 46 $C_2$ to $C_{24}$ saturated and unsaturated fatty acids (as listed in Tables 5–7) and 9 organic acids of the tri-carboxylic acid cycle (glycolic, lactic, malic, succinic, fumaric, citric, isocitric, pyruvic and α-ketoglutaric acid), which were obtained from Sigma-Aldrich or from Cayman Chemical (Arbor, Michigan, USA), all the targeted organic acids was prepared in methanol at 200 nmol/mL for each compound. This solution was further diluted step by step at a dilution ratio of 1 to 5 (v/v) to have standard solutions of S1 to S10.

Ten μL of each plasma was mixed with 90 μL of methanol. After vortex-mixing for 15 s, 3-min sonication and centrifugal clarification for 15 min, 50 μL of the supernatant, or 50 μL of each standard solution was mixed with 20 μL of a solution containing deuterium-labeled analogues of C8 to C24 even-carbon saturated fatty acids as internal standard, 25 μL of 200-mM 3-NPH solution and 2 μL of 150-mM EDC solution. The mixture was allowed to react at 35 ˚C for 60 min in a thermomixer at a shaking frequency of 900 rpm. After the reaction, 10 μL was injected onto a C8 UPLC column (2.1 mm I.D. x 100 mm, 1.7 μm) for LC separation with a mobile phase composed of 1 mM ammonium fluoride in water and isopropanol-acetonitrile for binary-solvent gradient elution at 0.4 mL/min and 55 ˚C. The efficient gradient was 10% to 100% in 14 min.

Linear-regression calibration curves were constructed using analyte-to-internal standard peak area ratios (As/Ai) versus molar concentrations (nmol/mL) of the standard solutions for each compound. Concentrations of organic acids detected in each sample were calculated from the standard curves of the individual analytes with internal calibration.

**Analysis of carnitines.** Quantitation of carnitines was carried out according to a previously published method by Han et al [34].

A mixed standard solution containing reference substances of 28 free and acyl carnitines, as previously described [34], was prepared in methanol at 10 nmol/mL for each compound. This solution was further diluted step by step at a dilution ratio of 1 to 4 (v/v) to have standard solutions of S1 to S9.

Fifteen μL of each plasma was mixed with 85 μL of methanol. After vortex-mixing for 30 s and sonication for 3 min, followed by centrifugal clarification at 15,000 rpm for 15 min, 50 μL of the supernatant was mixed with 50 μL of 100 mM of 3-NPH solution and 50 μL of a mixed solution containing 100 mM of EDC, HCl and 3% pyridine, all in 75% aqueous methanol. The mixture was allowed to react at 30 ˚C for 30 min in a thermomixer at a shaking frequency of

900 rpm. After the reaction, 50 μL of $^{13}C_6$-3NPH derivatives of all the targeted carnitines, which was in advance prepared in a "one-pot" reaction was added. After mixing, 20 μL of each solution was injected for quantitation of free and acyl carnitines in the samples by UPLC-MRM/MS with positive-ion detection using the previously described method [34]. A C8 UPLC column (2.1 x 100 mm, 1.8 μm) was used for LC separation with water-0.1% formic acid and acetonitrile as the mobile phase for binary gradient elution.

Linear-regression calibration curves were constructed using analyte-to-internal standard peak area ratios (As/Ai) versus molar concentrations (nmol/mL) of the standard solutions for each bile acid. Concentrations of carnitines in each sample were calculated from the calibration curves of the individual carnitines. Concentrations of organic acids detected in each sample were calculated from the standard curves of the individual analytes with internal calibration.

**Analysis of sphingolipids.** A mixed standard solution containing reference substances of 60 sphingolipids (from Avantis Polar Lipids, Alabaster, AL, USA; or Cayman chemical) was dissolved in methanol at 50 nmol/mL for each compound. This solution was further diluted step by step at a dilution ratio of 1 to 4 (v/v) to have standard solutions of S1 to S10.

20 μL of plasma was mixed with 180 μL of methanol-chloroform (1:1). After vortex-mixing for 30 s and sonication for 15 min at 15000 rpm. The supernatant was taken out and dried down in a nitrogen evaporator. The residue was resuspended in 50 μL of methanol.

10 μL of each standard solution or each sample solution was injected onto a C8 UPLC column (2.1 x 50 mm, 1.7 μm) for UPLC-MRM/MS with positive-ion detection, with 0.1% formic acid in water (A) and acetonitrile-isopropanol (1:1) (B) as the mobile phase for binary gradient elution, at 0.4 mL/min and 55 ˚C. The efficient gradient is 50% to 100% B in 15 min.

Linear-regression calibration curves were constructed using peak areas versus molar concentrations (nmol/mL) of the standard solutions for each sphingolipid. Concentrations of each detected lipid were calculated from the calibration curves with peak areas. For those sphingolipids detected but without the standard substances available, their concentrations were estimated using the calibration curves from one of the homologues in each sphingolipid class with the closest carbon number of their acyl chains.

**Analysis of glycerolipids, glycerophospholipids, and cholesteryl esters.** For these analytes, a metabolomics kit (Absolute IDQ p400 HR kit, Biocrates Life Sciences AG, Innsbruck, Austria) was used according to the manufacturer standard operating procedure using UHPLC with a C18 column coupled to a QExactive HF OrbiTrap mass spectrometer (Thermo Fisher Scientific, Waltham, MA, USA) with protocols previously published by the University of Alberta—The Metabolomics Innovation Centre [35, 36].

Only lipid metabolites were reported. The kit panel overlapped with the other targeted panels for carnitines and some sphingolipids. Only the analytes not measured by other targeted techniques were reported.

## Untargeted lipidomics

A Dionex Ultimate 3000 UHPLC system coupled to a Thermo Scientific LTQ-Orbitrap Velos Pro mass spectrometer equipped with an electrospray ionization (ESI) source was used.

Fifty μL of plasma was aliquoted to a 1.5-mL Eppendorf safe-lock tube. 250 μL of mixed methanol/chloroform (3:1) was added. The tube was vortex mixed for 20 s at 3000 rpm, sonicated in an ice-water bath for 3 min and then centrifuged at 15000 rpm for 20 min. The clear supernatant was taken out and transferred to an LC injection micro-vial. 10-μL aliquot were injected to run reversed-phase LC-MS for detection and relative quantitation of lipids, in (+) and (-) ion modes, respectively, with two rounds of LC-MS runs.

UHPLC-MS/MS runs were carried out for analysis of various lipids with the use of a C8 UHPLC column (2.1 x 50 mm, 1.7 μm) for chromatographic separation. The mobile phase was (A) 0.02% formic acid in water and (B) 0.02% formic acid in acetonitrile-isopropanol (1:1, v/v). The efficient gradient was 5% to 50% B in 5 min; 50% to 100% B in 15 min and 100% B for 2.5 min before the column was equilibrated for 4 min at 5% B between injections. The column flow was 400 μL/min and the column temperature was maintained at 55 ˚C. The MS instrument was operated in the survey-scan mode with full-mass and high-resolution Fourier transform MS detection at a mass resolution of 60,000 FWHM @ m/z 200. The mass scan range was m/z 70 to 1800 for both positive-ion and negative-ion detection. Along with the MS data, MS/MS data was also acquired using collision induced dissociation (CID) with top 6 acquisitions.

Two MS full-mass detection datasets were acquired. To process these MS datasets, the raw data files were converted to a common data format and were processed with XCMS (https://xcmsonline.scripps.edu/) in R for peak detection, retention time shift corrections, peak grouping and peak alignment. Mass de-isotoping and removal of chemical background noise peaks were performed, with partial manual interventions based on several rules in chemistry and mass spectrometry.

Lipid annotations were performed manually using the LIPIDMAPS structural database (LMSD) bulk structures [37] and LipidBlast software (Lipidblast-mz-lookup-v49 module) [38] for positive mode [(M+H)+ and (M+Na)+ ions] and for negative mode [(M-H)- ions]. A m/z error of 5ppm (Lipidmaps) or 0.008 Da (Lipidblast) was used. Only biologically relevant species were included in the search. LipidBlast was used to query glycerolipids (DG, TG), glycerophospholipids (PC, PE, PS, PG, PI, PA) and sphingolipids (SM, gangliosides, ceramides). LMSD was used to additionally query MG, cholesteryl esters (CE), and fatty acyls (search restricted to fatty acids and acyl-carnitines). Duplicate hits were removed manually. The LIPIDMAPS nomenclature was used for lipid names, category names, and abbreviations. Lipid species were reported at the brutto level of information [29].

## Statistical analysis

Targeted lipidomics data were reported with their median, interquartile range (IQR), minimum, and maximum for each lipid group and relative percentage for some lipid groups. A global barplot on polar coordinate was performed with all targeted panels to have a graphical representation of the whole lipidome. For untargeted lipidomics data, only the main lipid species per lipid groups were reported with the most common lipid species based on peak intensity.

For differences between sexes, targeted lipidomics data were pre-processed using auto scaling (mean centered and divided by standard deviation) for multivariate analysis. Metabolites below the limits of detection were removed from the analysis. Differences in individual lipid metabolites between sexes were then assessed using serial t-tests with a false discovery rate of 5%. The lipidome was explored using principal component analysis (PCA) to detect sex lipidomics signatures using an unsupervised technique. Lipid analytes contribution to respective principal components was assessed by evaluation of the PCA biplots. Differences between sexes were further explored using a supervised classifying multivariate tool: partial least squares—linear discriminant analysis (PLS-DA). Loading plots were also inspected for important features as well as the variable importance in projection (VIP) scores. A heatmap with hierarchical clustering was also generated for data exploration using the 50 most important analytes based on t-test p-values on normalized data. R (version 4.0.0, R foundation for statistical computing, Vienna, Austria) was used for descriptive statistical analysis, ggplot2 for bar plots [39], and MetaboAnalyst 4.0 for multivariate analysis and graphs [40].

# Results

## Nutritional analysis

The parrot pelleted diet contained 11.5% total fat including 1.6% of saturated fatty acids, 4.8% of monounsaturated fatty acids, and 5.1% of polyunsaturated fatty acids. The fatty acid profile included palmitic acid (FA 16:0) as the preponderant saturated fatty acid, oleic acid (FA 18:1) as the preponderant monounsaturated fatty acid, and linoleic acid (FA 18:2) as the preponderant polyunsaturated fatty acid. The combination of linoleic acid and oleic acid represented more than 80% of total dietary fatty acids. The complete fatty acid profile is reported in Table 1.

**Table 1. Fatty acid composition of the Quaker parrot pelleted diet.**

| Parameter | % of total fatty acids |
| --- | --- |
| FA(6:0) Caproic acid | <0.1 |
| FA(8:0) Caprylic acid | <0.1 |
| FA(10:0) Capric acid | <0.1 |
| FA(12:0) Lauric acid | <0.1 |
| FA(14:0) Myristic acid | <0.1 |
| FA(14:1) Myristoleic acid | <0.1 |
| FA(15:0) Pentadecylic acid | <0.1 |
| FA(15:1) Pentadecanoic acid | <0.1 |
| FA(16:0) Palmitic acid | 9.5 |
| FA(16:1) Palmitoleic acid | <0.1 |
| FA(17:0) Heptadecanoic acid | <0.1 |
| FA(17:1) Cis-10 heptadecenoic acid | <0.1 |
| FA(18:0) Stearic acid | 2.6 |
| FA(18:1) Elaidic acid | <0.1 |
| FA(18:1) Oleic acid | 41.3 |
| FA(18:2) Trans-linolelaidic acid | <0.1 |
| FA(18:2) Linoleic acid | 39.6 |
| FA(18:3) Gamma-linolenic acid | <0.1 |
| FA(18:3) Alpha-linolenic acid | 4.8 |
| FA(18:4) stearidonic acid | <0.1 |
| FA(20:0) Arachidic acid | 0.5 |
| FA(20:1) Eicosenoic acid | 0.6 |
| FA(20:2) Eicosadienoic acid | <0.1 |
| FA(20:3) Cis-Eicosatrienoic acid | <0.1 |
| FA(20:4) Arachidonic acid | <0.1 |
| FA(20:5) Eicosapentaeonic acid | <0.1 |
| FA(21:0) Heneicosanoic acid | <0.1 |
| FA(22:0) Behenic acid | 0.7 |
| FA(22:1) Erucic acid | <0.1 |
| FA(22:2) Cis-docosadienoic acid | <0.1 |
| FA(22:4) Docosatetraenoic acid | <0.1 |
| FA(22:5) Docosapentaenoic acid | <0.1 |
| FA(22:6) Docosahexaneoic acid | <0.1 |
| FA(24:0) Lignoceric acid | 0.3 |
| FA(24:1) Nervonic acid | <0.1 |

Limits of quantitation is 0.1%.

**Table 2. Number of lipid species or group of isomeric species quantified by targeted lipidomics panels in Quaker parrots (*Myiopsitta monachus*) plasma.**

| Category | Main class | Subclass | N |
|---|---|---|---|
| Fatty acyls | Fatty acids (FA) and conjugates (non-esterified) | Hydroxy fatty acids | 1 |
| | | Saturated fatty acids | 27 |
| | | Unsaturated fatty acids | 18 |
| | Fatty esters | Fatty acyl carnitines | 39 |
| Glycerolipids | Diradylglycerols | Acyl-alkylglycerols | 3 |
| | | Diacylglycerols (DG) | 14 |
| | Triradylglycerols | Triacylglycerols (TG) | 38 |
| Glycerophospholipids | Glycerophosphocholines | Monoalkylglycerophosphocholines | 4 |
| | | Monoacylglycerophosphocholines | 13 |
| | | Alkyl-acylglycerophosphocholines | 35 |
| | | Diacylglycerophosphocholines | 71 |
| Sphingolipids | Sphingoid bases (non-esterified) | Various subclasses | 9 |
| | Ceramides | N-acylsphinganines | 13 |
| | | N-acylsphingosines | 22 |
| | Phosphosphingolipids | Ceramide phosphocholines (sphingomyelins) | 17 |
| | Neutral glycosphingolipids | Simple GLc series | 9 |
| Sterol lipids | Bile acids and derivatives | Various subclasses | 75 |
| | Sterols | Cholesterol and derivatives | 10 |
| | | Sterol esters | 14 |
| Total | | | 432 |

## Plasma lipidome

The number of lipid species (not accounting for isomeric species, which are numerous for triacylglycerols and diaryl lipids) quantified by targeted lipidomics panels are reported in Table 2 and included 432 lipids. The highest number of lipids analyzed were for glycerophospholipids, but only included glycerophosphocholines.

A global representation of the targeted lipidomics panel is also presented in Fig 1.

The Quaker parrot plasma lipidome, as assessed using these incomplete panels, was dominated by sterol lipids and glycerophospholipids on a molar basis followed by glycerolipids (Fig 1 and Table 3). Cholesterol and its esters were the most abundant lipid species by far especially for free cholesterol and cholesteryl linoleate [CE(18:2)].

Untargeted lipidomics results yielded a high number of lipid species (Table 4). They included lipid categories and species not represented or measured in the targeted lipidomics panels. The most diverse lipid category was the glycerophospholipids, which was dominated by glycerophosphoethanolamines (PE) and glycerophosphocholines (PC) species. Of those, only PCs were quantified using targeted methods. Identified abundant species did not necessarily correspond to lipid found to be abundant on targeted panels.

Raw data for untargeted lipidomics prior to lipid identification and results of the targeted panels were published in the public domain in a permanent scientific data repository (Beaufrère H, 2020, The plasma lipidome of the Quaker parrot (*Myiopsitta monachus*), https://doi.org/10.5683/SP2/XUW31U, Scholars Portal Dataverse, V1).

**Fatty acyls.** Measured free fatty acids included non-esterified saturated fatty acids (Table 5), unsaturated fatty acids (Table 6) of short, medium, long, and very-long chains as well as one hydroxy-fatty acid (Table 7). Non-esterified fatty acids correspond to only a small proportion of all fatty acids in the plasma as most are esterified to various head groups.

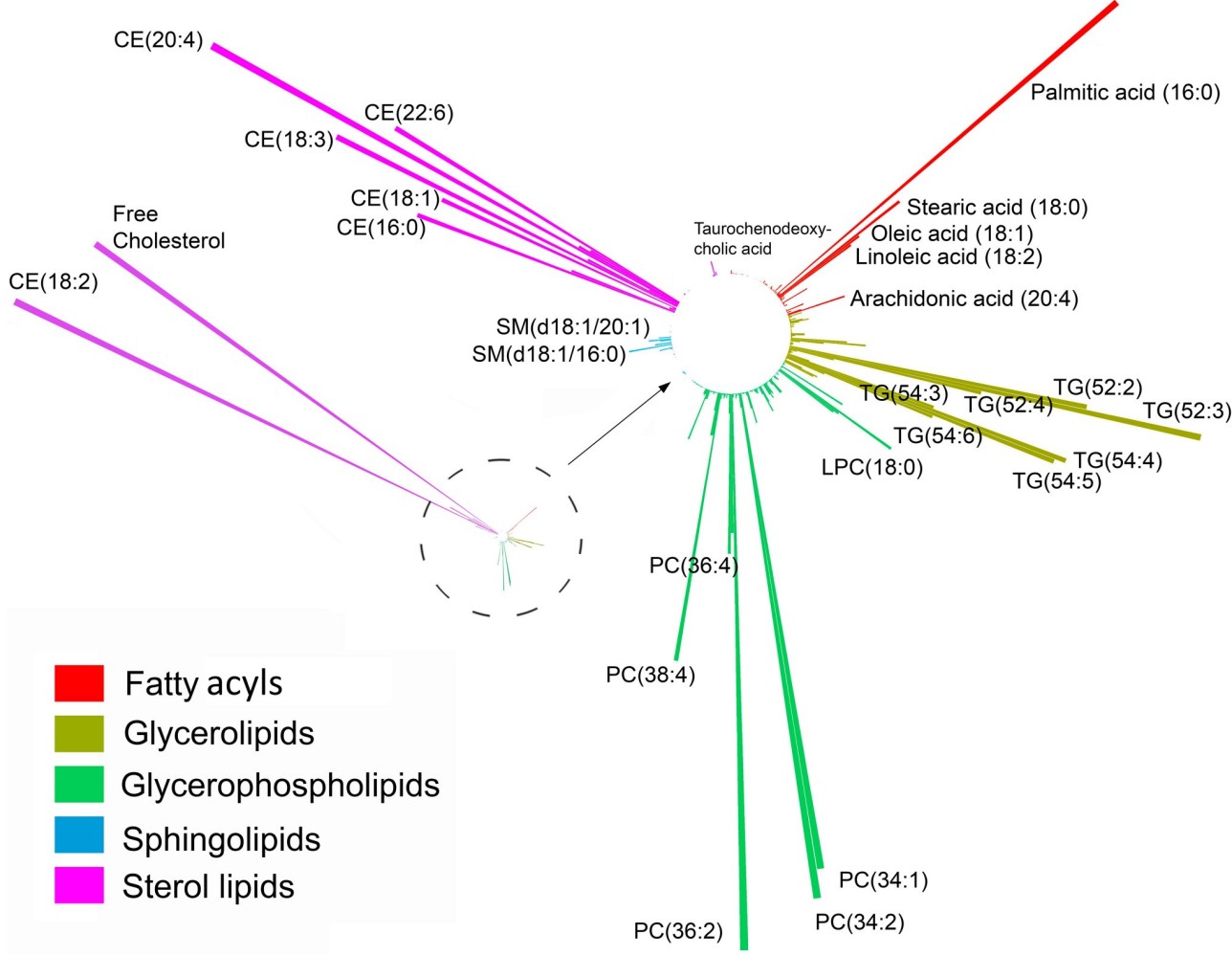

**Fig 1. Circular barplot of the mean concentration of lipid species across 5 lipid categories measured by targeted lipidomics in Quaker parrots (*Myiopsitta monachus*).** The dashed circle and arrow indicate the zoomed-in portion of the left barplot. Each lipid category is color coded differently. Lipid species abbreviation follows the LIPIDMAPS nomenclature. While lipid species were quantitively measured, comparisons across lipid category should be made carefully.

Palmitic acid was the most abundant free fatty acid in the plasma by far followed by stearic acid, oleic acid, linoleic acid, and arachidonic acid (Fig 1). It was also the most abundant free saturated fatty acid representing more than 68% of all plasmatic free saturated fatty acids. Palmitic and stearic acids represented 93% of all saturated fatty acids. Oleic acid was the most

**Table 3. Percentage of lipids by lipid category on a molar basis in the plasma of Quaker parrots (*Myiopsitta monachus*) as determined by targeted lipidomics panels.**

| Lipid category | % |
|---|---|
| Fatty acyls | 4.8 |
| Glycerolipids | 11.3 |
| Glycerophospholipids | 14.5 |
| Sphingolipids | 0.6 |
| Sterol lipids | 68.7 |

**Table 4. Number of lipid species or group of isomeric/isobaric species identified by untargeted lipidomics in six Quaker parrots (*Myiopsitta monachus*) plasma.** Lipid identification was performed using LIPIDMAPS structural database (LMSD) bulk structure and LipidBlast. The most abundant identified lipids are also displayed by category. Lipid species abbreviation follows the LIPIDMAPS nomenclature. Abundant species were determined based on peak intensity within lipid groups.

| Category | N | Abundant species |
|---|---|---|
| Fatty acyls | | |
| Fatty acids and conjugates (FA) | 161 | FA(22:6),FA(18:3),FA(20:4),FA(18:2),FA(18:1) |
| Fatty acyl carnitines (C) | 30 | C22:6,C18:1 |
| Glycerolipids | | |
| Monoacylglycerols (MG) | 8 | MG(18:1),MG(18:2) |
| Diacylglycerols (DG) | 0 | |
| Triacylglycerols (TG) | 57 | TG(52:2),TG(52:3),TG(50:1),TG(54:6),TG(54:5) |
| Glycerophospholipids | | |
| Glycerophosphates (PA) | 118 | PA(36:2),PA(34:1),PA(38:2),PA(36:1) |
| Glycerophosphocholines (PC) | 136 | LPC(18:2), PC(34:1),PC(34:2), LPC(20:4) |
| Glycerophosphoethanolamines (PE) | 211 | PE(36:2),PE(P-18:0) |
| Glycerophosphoglycerols (PG) | 65 | PG(36:2),PG(34:1),PG(36:0) |
| Glycerophosphoinositols (PI) | 61 | PI(36:2),PI(38:3),PI(34:2),PI(36:1),PI(34:1) |
| Glycerophosphoserines (PS) | 120 | PS(21:0),PS(35:1),PS(44:8),PS(40:6) |
| Sphingolipids | | |
| Ceramides (Cer) | 55 | Cer(36:2),Cer(24:1),Cer(26:0) |
| Sphingomyelins (SM) | 7 | SM(42:1) |
| Glycosphingolipids | 5 | [glycan]-Cer(38:1),[glycan]-Cer(34:1), |
| Sterol lipids | | |
| Cholesteryl esters (CE) | 5 | CE(22:6), CE(19:0),CE(18:2) |
| TOTAL | 1039 | |

common monounsaturated fatty acid, but also the most common unsaturated fatty acid. Linoleic acid was the most abundant polyunsaturated fatty acid and omega-6 fatty acid and docosahexaenoic acid (DHA) the most common omega-3 fatty acid.

Acyl-carnitines were also measured (Table 8). They are activated fatty acids that can be transported across mitochondrial membranes for beta-oxidation to produce energy. Unsurprisingly, acetyl carnitine was the most common acyl carnitines as it facilitates the movement of acetyl-CoA as end-products of mitochondrial fatty acid oxidation. Oleyl-carnitine and linoleyl-carnitine were the most abundant long-chain fatty acyl carnitines.

Fatty acyls also include a large number of lipid mediators, which were also measured in these parrots, but as part of a different study on the plasma mediator lipidome.

**Glycerolipids.** The most abundant glycerolipids were triacylglycerols (Table 9). They were quantified based on identification at the sum composition. The most common species were TGs in C52 or C54 with 2 to 6 double bonds. The four most common TG species represented more than half (57.3%) of all TGs on a molar basis. Untargeted lipidomics also identified these TGs as common. While only the sum composition was used, based on typical stereospecific structures of biological TGs in animals and common plasma free fatty acids from Tables 5 and 6, they likely contained a high proportion of palmitic acid, and C18 fatty acids such as stearic, oleic, linoleic, and alpha-linolenic acids and arachidonic acid as a highly unsaturated fatty acid.

Diacylglycerols (Table 10) and some ether-linked diarylglycerols (Table 11) were quantified and were present at substantially lower concentrations than TGs. They are metabolic intermediates and metabolites of TGs and glycerophospholipids. DGs in C36 were the most common,

**Table 5. Plasma free saturated fatty acid concentration (µM) in six Quaker parrots (*Myiopsitta monachus*) determined by mass spectrometry.**

| Lipids | Median | IQR | Min | Max | % |
|---|---|---|---|---|---|
| FA(02:0) Acetic acid | 5.234 | 2.191 | 1.013 | 6.978 | 0.5 |
| FA(03:0) Propionic acid | 0.019 | NA | <LOD | 0.348 | 0.0 |
| FA(04:0) Butyric acid | 1.271 | 0.787 | 0.912 | 2.19 | 0.1 |
| FA(04:0) Isobutyric acid | 1.52 | 0.248 | 1.133 | 1.88 | 0.1 |
| FA(05:0) 2-Methylbutyric acid | 1.965 | 0.306 | 1.67 | 2.48 | 0.2 |
| FA(05:0) Isovaleric acid | 0.864 | 0.178 | 0.62 | 1.458 | 0.1 |
| FA(05:0) Valeric acid | 0.051 | 0.135 | 0 | 0.195 | 0.0 |
| FA(06:0) 3-Methylvaleric acid | 0.04 | 0.033 | 0 | 0.073 | 0.0 |
| FA(06:0) Caproic acid | 0.357 | 0.227 | 0.186 | 0.515 | 0.0 |
| FA(06:0) Isocaproic acid | <LOD | NA | <LOD | 0.116 | 0.0 |
| FA(08:0) Caprylic acid | 4.843 | 1.517 | 1.325 | 5.801 | 0.4 |
| FA(09:0) Pelargonic acid | 1.666 | 0.512 | 0.814 | 2.256 | 0.2 |
| FA(10:0) Capric acid | 1.172 | 0.399 | 0.918 | 1.703 | 0.1 |
| FA(11:0) Undecylic acid | 0.19 | 0.029 | 0.096 | 0.21 | 0.0 |
| FA(12:0) Lauric acid | 2.779 | 0.599 | 2.319 | 3.443 | 0.3 |
| FA(13:0) Tridecylic acid | 0.062 | 0.15 | 0.017 | 0.249 | 0.0 |
| FA(14:0) Myristic acid | 17.871 | 3.756 | 13.139 | 20.906 | 1.6 |
| FA(15:0) Pentadecylic acid | 2.918 | 0.855 | 2.279 | 3.428 | 0.3 |
| FA(16:0) Palmitic acid | 744.897 | 75.048 | 472.605 | 1014.033 | 68.6 |
| FA(17:0) Margaric acid | 5.97 | 1.271 | 3.534 | 7.728 | 0.5 |
| FA(18:0) Stearic acid | 264.991 | 23.737 | 162.998 | 311.587 | 24.4 |
| FA(19:0) Nonadecylic acid | 2.699 | 0.492 | 2.483 | 3.702 | 0.2 |
| FA(20:0) Arachidic acid | 8.135 | 1.556 | 6.504 | 9.069 | 0.7 |
| FA(21:0) Heneicosylic acid | 0.867 | 0.254 | 0.505 | 0.964 | 0.1 |
| FA(22:0) Behenic acid | 6.088 | 1.776 | 3.312 | 7.097 | 0.6 |
| FA(23:0) Tricosylic acid | 0.87 | 0.345 | 0.577 | 1.371 | 0.1 |
| FA(24:0) Lignoceric acid | 8.134 | 1.426 | 4.36 | 9.473 | 0.7 |

IQR, interquartile range; NA, not applicable; LOD, limits of detection.

likely containing either a combination of C18 fatty acids (stearic, oleic, linoleic, linolenic) or a combination of palmitic acid and arachidonic acid. DGs were identified at the sum composition level of information and it is unknown whether they were 1,2 DG or 1,3 DG. They were not detected through untargeted lipidomics.

Monoacylglycerols were not quantified by targeted techniques, but untargeted methods tentatively identified MG(18:1) and MG(18:2) as common plasmatic species.

**Glycerophospholipids.** Only glycerophosphocholines were quantified as part of targeted lipidomics panels, but untargeted methods detected a large number of other glycerophospholipids (Table 4). Glycerophosphocholines (PCs) was one of the most diverse lipid categories and also included the highest number of lipid species quantified by targeted panels (Tables 12–14). Phosphatidylcholines were the most common PCs especially the species in C34, C36 and C38. As they only contain 2 fatty acid chains, the most likely components were palmitic acid, stearic acid, linoleic acid, linolenic acid, and arachidonic acid. Based on untargeted lipidomics, PE are also common and abundant species in parrot plasma with PE(36:2) common [as for PC(36:2)], but those could not be confirmed and further quantified by targeted techniques.

Ether-linked PCs (PC-O) were also common and quantified (Table 13).

**Table 6. Plasma free unsaturated fatty acid concentration (μM) in six Quaker parrots (*Myiopsitta monachus*) determined by mass spectrometry.**

| Lipids | Median | IQR | Min | Max | % |
|---|---|---|---|---|---|
| FA(14:1) Myristoleic acid | 0.506 | 0.088 | 0.396 | 0.635 | 0.1 |
| FA(16:1) Palmitoleic acid | 6.071 | 1.638 | 4.186 | 9.562 | 1.4 |
| FA(18:1) Oleic acid | 173.823 | 35.534 | 92.953 | 234.033 | 40.0 |
| FA(18:2) Linoleic acid | 157.261 | 40 | 76.826 | 210.138 | 36.2 |
| FA(18:3) α-Linolenic acid | 8.921 | 2.776 | 6.122 | 12.686 | 2.1 |
| FA(20:1) Gondoic acid | 1.524 | 0.397 | 1.251 | 1.95 | 0.4 |
| FA(20:2) Eicosadienoic acid | 0.474 | 0.143 | 0.324 | 0.637 | 0.1 |
| FA(20:3) Dihomo-γ-linolenic acid | 0.875 | 0.255 | 0.633 | 1.158 | 0.2 |
| FA(20:4) Arachidonic acid | 47.954 | 11.11 | 38.127 | 54.582 | 11.0 |
| FA(20:5) Eicosapentaenoic acid (EPA) | 1.76 | 0.635 | 1.381 | 2.583 | 0.4 |
| FA(20:6) Eicosahexaenoic acid | 1.891 | 0.628 | 1.459 | 2.643 | 0.4 |
| FA(22:1) Erucic acid | 0.17 | 0.036 | 0.099 | 0.184 | 0.0 |
| FA(22:2) Docosadienoic acid | 0.069 | 0.009 | 0.047 | 0.088 | 0.0 |
| FA(22:3) Docosatrienoic acid | 0.078 | NA | <LOD | 0.324 | 0.0 |
| FA(22:4) Docosatetraenoic acid | <LOD | NA | <LOD | <LOD | 0.0 |
| FA(22:5) Docosapentaenoic acid | 1.645 | 0.369 | 1.21 | 1.856 | 0.4 |
| FA(22:6) Docosahexaenoic acid (DHA) | 31.145 | 6.36 | 22.717 | 41.09 | 7.2 |
| FA(24:1) Nervonic acid | 0.16 | 0.04 | 0.105 | 0.189 | 0.0 |

IQR, interquartile range; NA, not applicable; LOD, limits of detection.

Lysophophatidylcholines (LPCs) are reported in Table 4 along with LPC-O and are metabolites of PCs. LPCs in C18 and with palmitic acid were the most abundant.

**Sphingolipids.**   Most sphingolipids had sphingosine (d18:1) as the sphingoid base; however only sphingolipids with sphingosine or sphinganine (d18:0) were quantified by targeted methods and only the sum composition was obtained for untargeted methods. Sphingosine was also the most abundant non-esterified sphingoid base in the plasma at 87.2% (Table 15).

Sphingomyelins (SM) were the most abundant plasma sphingolipids and sphingomyelin with long-chain saturated fatty acids were the preponderant species such as the most common, palmitoyl sphingomyelin [SM(d18:1/16:0)]. Saturated fatty acid sphingomyelins accounted for 81.8% of all SMs (Table 16).

Ceramides and dihydroceramides were in much lower concentrations in the plasma than SMs at about 3% of all measured sphingolipids. There was a tendency for these lipids to have very long chain fatty acids (>C22) as esterified fatty acids. The most common ceramide included nervonic acid [Cer(d18:1/24:1)] (Table 17). Dihydroceramides were in minute concentrations when compared to ceramides (Table 18).

A few cerebrosides (monoglycosylceramides) and globosides (polyglycosylceramides) were quantified (Table 19). Like ceramides and dihydroceramides, they had very long chain fatty acids.

**Table 7. Plasma free hydroxy fatty acid concentration (μM) in six Quaker parrots (*Myiopsitta monachus*) determined by mass spectrometry.**

| Lipids | Median | IQR | Min | Max |
|---|---|---|---|---|
| FA(16:0-OH) hydroxypalmitic acid | 0.1 | 0.057 | 0.064 | 0.2 |

IQR, interquartile range.

**Table 8. Plasma acyl carnitines concentration (µM) in six Quaker parrots (*Myiopsitta monachus*) determined by mass spectrometry.**

| Lipids | Median | IQR | Min | Max | % |
|---|---|---|---|---|---|
| C02:0 Acetyl carnitine | 5.77 | 1.512 | 3.25 | 6.78 | 45.6 |
| C03:0 Malonyl-carnitine | 0.411 | 0.113 | 0.291 | 0.552 | 3.2 |
| C03:0 Propionyl-carnitine | 0.347 | 0.074 | 0.283 | 0.433 | 2.7 |
| C04:0 Butyryl-carnitine | 0.578 | 0.148 | 0.356 | 0.681 | 4.6 |
| C04:0 Hydroxybutyryl-carnitine | 0.107 | 0.055 | 0.039 | 0.138 | 0.8 |
| C04:0 Isobutyryl—Carnitine | 0.212 | 0.072 | 0.136 | 0.257 | 1.7 |
| C04:0 Methylmalonyl-carnitine | 0.791 | 0.266 | 0.663 | 0.96 | 6.2 |
| C04:0 Succinyl-carnitine | 0.042 | 0.024 | 0.019 | 0.065 | 0.3 |
| C05:0 1-3-Methylcrotonyl-L-Carnitine | 0.004 | 0.002 | 0.003 | 0.005 | 0.0 |
| C05:0 2-Methylbutyryl-L-Carnitine | 0.326 | 0.089 | 0.261 | 0.44 | 2.6 |
| C05:0 3-hydroxyisovaleryl-carnitine | 0.199 | 0.048 | 0.132 | 0.311 | 1.6 |
| C05:0 Glutaryl-carnitine | 0.328 | 0.057 | 0.277 | 0.404 | 2.6 |
| C05:0 Isovaleryl-Carnitine | 0.346 | 0.05 | 0.288 | 0.4 | 2.7 |
| C05:0 Valeryl-carnitine | 1.4 | 0.325 | 0.625 | 1.62 | 11.1 |
| C05:1 Tiglyl-carnitine | 0.034 | 0.009 | 0.026 | 0.043 | 0.3 |
| C06:0 Hexanoyl-carnitine | 0.056 | 0.034 | 0.027 | 0.084 | 0.4 |
| C06:0 3-Methylglutaryl-Carnitine | 0.003 | 0.001 | 0.002 | 0.004 | 0.0 |
| C06:1 Hexenoylcarnitine | 0.02 | 0.018 | 0.01 | 0.086 | 0.2 |
| C08:0 Octanoyl-carnitine | 0.148 | 0.107 | 0.051 | 0.427 | 1.2 |
| C08:1 Hydroxyoctenoyl-carnitine | 0.444 | 0.592 | 0.153 | 1.11 | 3.5 |
| C08:1 Octenoyl-carnitine | 0.103 | 0.083 | 0.009 | 0.164 | 0.8 |
| C10:0 Decanoyl-carnitine | 0.024 | 0.007 | 0.015 | 0.028 | 0.2 |
| C10:1 Decenoyl-carnitine | 0.024 | 0.008 | 0.012 | 0.03 | 0.2 |
| C12:0 Dodecanedioyl-carnitine | 0.01 | 0.005 | 0.001 | 0.012 | 0.1 |
| C12:0 Dodecanoyl-carnitine | 0.03 | 0.007 | 0.025 | 0.035 | 0.2 |
| C12:1 Dodecenoyl-carnitine | 0.017 | 0.006 | 0.014 | 0.021 | 0.1 |
| C14:0 Tetradecanoyl-carnitine | 0.029 | 0.01 | 0.023 | 0.038 | 0.2 |
| C14:1 Tetradecenoylcarnitine | 0.066 | 0.02 | 0.042 | 0.079 | 0.5 |
| C14:2 Hydroxytetradecadienoyl-carnitine | 0.017 | 0.004 | 0.011 | 0.019 | 0.1 |
| C14:2 Tetradecadienoyl-carnitine | 0.035 | 0.011 | 0.021 | 0.04 | 0.3 |
| C16:0 3-hydroxyhexadecanoyl-carnitine | 0.005 | 0.002 | 0.003 | 0.007 | 0.0 |
| C16:0 Palmitoyl-carnitine | 0.118 | 0.04 | 0.091 | 0.18 | 0.9 |
| C16:1 Hexadecenoyl-carnitine | 0.009 | 0.005 | 0.006 | 0.015 | 0.1 |
| C16:2 Hexadecadienoyl-carnitine | 0.012 | 0.006 | 0.009 | 0.025 | 0.1 |
| C18:0 Octadecanoyl-carnitine | 0.074 | 0.023 | 0.053 | 0.109 | 0.6 |
| C18:1 Oleyl-carnitine | 0.294 | 0.105 | 0.192 | 0.403 | 2.3 |
| C18:2 Linoleyl-carnitine | 0.19 | 0.054 | 0.101 | 0.226 | 1.5 |
| C20:0 Arachidyl-carnitine | 0.03 | 0.008 | 0.025 | 0.036 | 0.2 |
| C20:4 Arachidonoyl-carnitine | 0.011 | 0.004 | 0.009 | 0.015 | 0.1 |

IQR, interquartile range.

**Sterol lipids.** As mentioned above, sterol lipids, in particular free cholesterol and CE (18:2) dominate the plasma lipidome of the Quaker parrot. Cholesteryl esters (CE) (Table 20) compose about 2/3 of plasma cholesterol with the remaining 1/3 being free cholesterol (Table 21). Most important CE had polyunsaturated fatty acids such as linoleic acid, linolenic acid, arachidonic acid, and DHA.

**Table 9. Plasma triacylglycerols concentration (μM) in six Quaker parrots (*Myiopsitta monachus*) determined by mass spectrometry.**

| Lipids | Median | IQR | Min | Max | % |
|---|---|---|---|---|---|
| TG(44:1) | 2.095 | 0.42 | 0.907 | 2.76 | 0.1 |
| TG(44:2) | 1.255 | 1.318 | 0.027 | 1.83 | 0.0 |
| TG(44:4) | 0.012 | 0.014 | 0.007 | 2.16 | 0.0 |
| TG(46:2) | 2.31 | 0.737 | 1.42 | 3.05 | 0.1 |
| TG(48:1) | 22.65 | 8.35 | 14.5 | 34.8 | 0.6 |
| TG(48:2) | 22.25 | 7.225 | 17.3 | 27.1 | 0.6 |
| TG(48:3) | 7.125 | 2.3 | 4.34 | 9.25 | 0.2 |
| TG(49:1) | 2.19 | 0.84 | 1.27 | 2.9 | 0.1 |
| TG(49:2) | 1.96 | 0.615 | 1.38 | 3.03 | 0.1 |
| TG(50:1) | 99.2 | 40.95 | 51.2 | 137 | 2.6 |
| TG(50:2) | 130.5 | 56.125 | 75.7 | 170 | 3.5 |
| TG(50:3) | 56.4 | 17.8 | 33.9 | 75.6 | 1.5 |
| TG(50:4) | 15.5 | 2.925 | 10.4 | 18.5 | 0.4 |
| TG(51:1) | 0.018 | 1.053 | 0.015 | 2.44 | 0.0 |
| TG(51:2) | 4.785 | 1.757 | 3 | 6.27 | 0.1 |
| TG(51:3) | 4.79 | 1.2 | 2.59 | 6.29 | 0.1 |
| TG(51:4) | 3.485 | 0.893 | 2.02 | 4.49 | 0.1 |
| TG(51:5) | 1.685 | NA | <LOD | 2.45 | 0.0 |
| TG(52:2) | 488.5 | 190.25 | 321 | 726 | 12.9 |
| TG(52:3) | 678 | 143 | 438 | 897 | 18.0 |
| TG(52:4) | 330 | 54.25 | 182 | 413 | 8.7 |
| TG(52:5) | 79.3 | 11.725 | 41 | 97.3 | 2.1 |
| TG(52:6) | 14.25 | 2.5 | 8.33 | 16.2 | 0.4 |
| TG(52:7) | 4.055 | 0.265 | 2.98 | 4.63 | 0.1 |
| TG(53:3) | 7.14 | 1.407 | 4.24 | 8.69 | 0.2 |
| TG(53:4) | 5.05 | 1.025 | 2.96 | 6.66 | 0.1 |
| TG(53:5) | 3.205 | 0.818 | 1.9 | 3.89 | 0.1 |
| TG(54:2) | 42 | 34.9 | 6.19 | 74.4 | 1.1 |
| TG(54:3) | 259 | 93.25 | 123 | 352 | 6.9 |
| TG(54:4) | 504.5 | 158 | 212 | 730 | 13.4 |
| TG(54:5) | 490 | 130.75 | 206 | 727 | 13.0 |
| TG(54:6) | 269.5 | 55 | 123 | 388 | 7.1 |
| TG(54:7) | 70.3 | 15.3 | 32.1 | 103 | 1.9 |
| TG(55:6) | 2.28 | 0.507 | 1.66 | 2.64 | 0.1 |
| TG(55:7) | 1.51 | 1.29 | 0.034 | 2.23 | 0.0 |
| TG(56:6) | 31.85 | 7.225 | 23.4 | 38 | 0.8 |
| TG(56:7) | 60.95 | 22.6 | 43.3 | 74.8 | 1.6 |
| TG(56:8) | 56.1 | 24.825 | 37.3 | 70.8 | 1.5 |

IQR, interquartile range; NA, not applicable; LOD, limits of detection.

While free cholesterol was the main non-esterified sterol, other species were quantified as well as some steroids, which are synthesized from cholesterol (Table 21). Corticosterone, the most important steroid of birds, was not part of this targeted steroid panel.

A comprehensive bile acid panel was also performed on the plasma of these parrots (Table 22). By far, the most common plasma bile acid of birds was taurochenodeoxycholic

**Table 10. Plasma diacylglycerols concentration (μM) in six Quaker parrots (*Myiopsitta monachus*) determined by mass spectrometry.**

| Lipids | Median | IQR | Min | Max | % |
|---|---|---|---|---|---|
| DG(32:1) | 1.375 | 0.403 | 0.85 | 2.2 | 1.4 |
| DG(32:2) | 1.365 | 0.638 | 0.934 | 2.03 | 1.4 |
| DG(34:1) | 9.865 | 3.043 | 7.69 | 13.4 | 10.1 |
| DG(34:3) | 2.315 | 0.757 | 1.45 | 3.09 | 2.4 |
| DG(36:2) | 20.4 | 5.325 | 15.9 | 28.4 | 20.8 |
| DG(36:3) | 31.95 | 9.75 | 21.7 | 40.1 | 32.6 |
| DG(36:4) | 14.65 | 4.45 | 9.1 | 20.8 | 14.9 |
| DG(38:0) | 0.061 | NA | <LOD | 4 | 0.1 |
| DG(38:5) | 6.675 | 1.51 | 4.61 | 7.23 | 6.8 |
| DG(39:0) | 1.745 | 0.565 | 0.929 | 2.08 | 1.8 |
| DG(41:1) | 6.12 | 1.365 | 3.45 | 6.73 | 6.2 |
| DG(42:1) | 0.976 | 0.253 | 0.677 | 1.12 | 1.0 |
| DG(42:2) | 0.613 | 0.347 | 0.582 | 1.18 | 0.6 |
| DG(44:3) | 0.003 | NA | <LOD | 2.29 | 0.0 |

IQR, interquartile range; NA, not applicable; LOD, limits of detection.

acid. Taurocholic acid was the second most common plasma bile acid and together with tauro-chenodeoxycholic acid represented 87.4% of all bile acids.

**Others.** Other metabolites associated with lipid metabolism were quantified as part of the targeted lipidomics panels. These included dicarboxylic acids important in the TCA cycle (which provides substrates for fatty acid and cholesterol biosynthesis), free carnitine and other lipid-like metabolites (Table 23).

## Sex differences

On serial t-tests for targeted results on lipid concentrations, only 2 lipid species, both minor PCs (see Tables 12 and 13), were found to be significantly different between sexes, PC(42:4) (q = 0.024) and PC-O(34:3) (q = 0.024). The sexes were well clustered on PCA and PLS-DA plots without overlap (Fig 2). The PCA biplot was difficult to interpret because of the number of variables. Using VIP scores, the most important discriminating variables between sexes on PLS-DA were mainly PCs [PC(42:4) and PC-O(34:3) being the most important], acyl-carnitines (C12:1, C14:0, and C12:0 being the most important), and TG(56:7) for females, and Cer (d18:1/18:0) for males. The PCA model explained 57.4% of the variance and the PLS-DA model explained 54.7% of the variance.

**Table 11. Plasma acylalkylglycerols concentration (μM) in six Quaker parrots (*Myiopsitta monachus*) determined by mass spectrometry.**

| Lipids | Median | IQR | Min | Max | % |
|---|---|---|---|---|---|
| DG-O(32:2) | 3.005 | 0.993 | 2.01 | 3.72 | 9.5 |
| DG-O(34:1) | 23 | 6.65 | 18 | 29.6 | 72.7 |
| DG-O(36:4) | 5.615 | 0.72 | 4.44 | 7.67 | 17.8 |

IQR, interquartile range; NA, not applicable; LOD, limits of detection.

**Table 12. Plasma diacylglycerophosphocholines (phosphatidylcholines) concentration (μM) in six Quaker parrots (*Myiopsitta monachus*) determined by mass spectrometry.**

| Lipids | Median | IQR | Min | Max | % |
|---|---|---|---|---|---|
| PC(30:0) | 1.6 | 0.235 | 1.29 | 1.97 | 0.0 |
| PC(31:0) | 0.282 | 0.109 | 0.223 | 0.424 | 0.0 |
| PC(31:1) | 0.076 | 0.037 | 0.034 | 0.117 | 0.0 |
| PC(32:0) | 53.55 | 10.425 | 39 | 67.2 | 1.3 |
| PC(32:1) | 32.65 | 17.075 | 12.6 | 39.1 | 0.8 |
| PC(32:2) | 5.51 | 0.885 | 4.35 | 5.95 | 0.1 |
| PC(32:6) | 0.943 | 0.232 | 0.564 | 0.994 | 0.0 |
| PC(33:0) | 1.019 | 0.193 | 0.782 | 1.33 | 0.0 |
| PC(33:1) | 1.95 | 0.762 | 1.31 | 2.36 | 0.0 |
| PC(33:2) | 1.655 | 0.245 | 1.54 | 1.99 | 0.0 |
| PC(33:4) | 0.216 | NA | <LOD | 1.26 | 0.0 |
| PC(34:1) | 865.5 | 187.25 | 519 | 976 | 20.2 |
| PC(34:2) | 893 | 138.5 | 684 | 945 | 20.9 |
| PC(34:3) | 23.9 | 7.3 | 16.2 | 30.8 | 0.6 |
| PC(34:4) | 2.1 | 0.865 | 1.33 | 2.63 | 0.0 |
| PC(35:0) | 0.209 | NA | <LOD | 0.962 | 0.0 |
| PC(35:1) | 4.145 | 1.28 | 2.91 | 4.94 | 0.1 |
| PC(35:2) | 8.17 | 1 | 6.18 | 9.23 | 0.2 |
| PC(35:3) | 0.838 | 0.187 | 0.512 | 0.944 | 0.0 |
| PC(35:4) | 0.46 | 0.109 | 0.345 | 0.78 | 0.0 |
| PC(36:1) | 6.255 | 3.42 | 1.4 | 8.37 | 0.1 |
| PC(36:2) | 978 | 211 | 749 | 1053 | 22.8 |
| PC(36:3) | 252.5 | 59 | 161 | 283 | 5.9 |
| PC(36:4) | 285.5 | 93.75 | 149 | 327 | 6.7 |
| PC(36:5) | 30.1 | 8.8 | 23.2 | 49.3 | 0.7 |
| PC(36:6) | 1.865 | 0.37 | 1.38 | 2.03 | 0.0 |
| PC(37:1) | 1.61 | 0.555 | 1.11 | 2 | 0.0 |
| PC(37:2) | 6.815 | 1.96 | 5.34 | 8.96 | 0.2 |
| PC(37:3) | 0.826 | 0.348 | 0.535 | 1.14 | 0.0 |
| PC(37:4) | 5.04 | 1.322 | 2.8 | 5.95 | 0.1 |
| PC(37:5) | 8.43 | 2.688 | 5.69 | 9.67 | 0.2 |
| PC(37:6) | 1.995 | 0.24 | 1.46 | 3.44 | 0.0 |
| PC(38:1) | 2.51 | 0.663 | 1.59 | 3.35 | 0.1 |
| PC(38:2) | 10.65 | 2.643 | 6.9 | 12 | 0.2 |
| PC(38:4) | 437.5 | 148.75 | 331 | 572 | 10.2 |
| PC(38:5) | 74.6 | 12.65 | 30.4 | 93.5 | 1.7 |
| PC(38:6) | 74 | 7.025 | 55.5 | 80.6 | 1.7 |
| PC(38:7) | 29.7 | 7.85 | 20.9 | 38.9 | 0.7 |
| PC(39:2) | 0.572 | 0.106 | 0.422 | 0.658 | 0.0 |
| PC(39:3) | 0.55 | 0.167 | 0.043 | 0.922 | 0.0 |
| PC(39:4) | 1.625 | 0.52 | 1.37 | 2.07 | 0.0 |
| PC(39:5) | 1.165 | 0.35 | 0.868 | 1.92 | 0.0 |
| PC(39:6) | 3.97 | 1.428 | 2.33 | 5.34 | 0.1 |
| PC(39:7) | 1.74 | 0.725 | 0.572 | 2.11 | 0.0 |
| PC(40:1) | 0.973 | 0.254 | 0.669 | 1.12 | 0.0 |
| PC(40:2) | 2.77 | 0.742 | 1.97 | 2.98 | 0.1 |

(*Continued*)

**Table 12.** (Continued)

| Lipids | Median | IQR | Min | Max | % |
|---|---|---|---|---|---|
| PC(40:3) | 0.281 | NA | <LOD | 1.2 | 0.0 |
| PC(40:4) | 15.4 | 2.275 | 11.2 | 18.3 | 0.4 |
| PC(40:5) | 16.7 | 3.6 | 12.6 | 19.6 | 0.4 |
| PC(40:6) | 91.85 | 17.525 | 62.6 | 110 | 2.1 |
| PC(40:7) | 7.645 | 11.863 | <LOD | 21.5 | 0.2 |
| PC(40:8) | 9.04 | NA | <LOD | 31.2 | 0.2 |
| PC(40:9) | 8.96 | 3.745 | 6.1 | 12.3 | 0.2 |
| PC(41:2) | 0.158 | NA | <LOD | 0.974 | 0.0 |
| PC(41:3) | 0.034 | NA | <LOD | 0.447 | 0.0 |
| PC(41:4) | 0.117 | 0.144 | 0.032 | 0.63 | 0.0 |
| PC(41:5) | 0.514 | 0.119 | 0.362 | 0.624 | 0.0 |
| PC(41:8) | 0.253 | NA | <LOD | 0.436 | 0.0 |
| PC(42:10) | 1.19 | NA | <LOD | 4.42 | 0.0 |
| PC(42:2) | 0.238 | NA | <LOD | 0.829 | 0.0 |
| PC(42:4) | 1.147 | 0.52 | 0.861 | 1.44 | 0.0 |
| PC(42:5) | 0.574 | 0.236 | 0.417 | 0.782 | 0.0 |
| PC(42:6) | 1.06 | 0.222 | 0.83 | 1.49 | 0.0 |
| PC(42:7) | 0.85 | 0.672 | 0.315 | 1.67 | 0.0 |
| PC(43:6) | 1.59 | 0.383 | 1.05 | 1.73 | 0.0 |
| PC(44:10) | 0.613 | NA | <LOD | 0.921 | 0.0 |
| PC(44:5) | 0.968 | 0.464 | 0.601 | 1.51 | 0.0 |
| PC(44:6) | 1.101 | 0.908 | 0.137 | 1.57 | 0.0 |
| PC(44:7) | 1.004 | 0.584 | 0.006 | 2.15 | 0.0 |
| PC(46:1) | 0.022 | 0.028 | 0.014 | 0.143 | 0.0 |
| PC(46:2) | 0.867 | 0.305 | 0.617 | 1.08 | 0.0 |

IQR, interquartile range; NA, not applicable; LOD, limits of detection.

The heatmap also suggested different lipidome profiles between sexes (Fig 3). Most lipids were in higher plasma concentrations in female parrots, in particular glycerophospholipids, acyl-carnitines, and some TGs and CEs. A few lipid species were higher in males, all being sphingolipids.

## Discussion

This report presents the first comprehensive database of plasma lipid species in a psittacine bird. The plasma lipidome of animals is astoundingly diverse and several quantitative or semi-quantitative mass spectrometric methods are typically needed to grasp a portion of this diversity due to the various molecular structures and abundance of the different lipid categories [21, 41, 42]. While the targeted lipidomics panels used here give a good overview of the plasma lipidome of the Quaker parrot, only a fraction of the plasma lipids were quantified. As some lipids are structurally complex, we only reported individual lipid species or group of species at the brutto level (sum composition of carbons and carbon-carbon double bonds) for most complex lipids such as glycerolipids and glycerophospholipids or medio level (with added knowledge of fatty acyl chains) for sphingolipids [29]. For complex lipids with many possible combination of fatty acids such as TG, a single reported species such as displayed in Table 9 may encompass several dozen TG isomeric species (same elemental composition, but different lipids), so this is

**Table 13. Plasma alkylacylglycerophosphocholines concentration (µM) in six Quaker parrots (*Myiopsitta monachus*) determined by mass spectrometry.**

| Lipids | Median | IQR | Min | Max | % |
|---|---|---|---|---|---|
| PC-O(26:0) | 0.083 | NA | <LOD | 0.157 | 0.0 |
| PC-O(30:0) | 0.197 | 0.004 | 0.159 | 0.231 | 0.1 |
| PC-O(32:0) | 8.745 | 2.088 | 8.61 | 12.2 | 3.2 |
| PC-O(32:1) | 5.605 | 1.628 | 5.19 | 7.72 | 2.0 |
| PC-O(32:2) | 0.309 | 0.091 | 0.241 | 0.419 | 0.1 |
| PC-O(33:2) | 0.595 | 0.04 | 0.478 | 0.841 | 0.2 |
| PC-O(33:3) | 0.43 | 0.086 | 0.367 | 0.497 | 0.2 |
| PC-O(34:0) | 2.215 | 0.678 | 1.94 | 2.96 | 0.8 |
| PC-O(34:1) | 25.15 | 8.175 | 23 | 36.1 | 9.2 |
| PC-O(34:2) | 25.25 | 9.2 | 23.3 | 36 | 9.2 |
| PC-O(34:3) | 9.18 | 1.892 | 8.17 | 10.3 | 3.3 |
| PC-O(34:4) | 0.414 | 0.42 | 0.318 | 1.2 | 0.2 |
| PC-O(35:3) | 0.503 | 0.763 | 0.112 | 1.85 | 0.2 |
| PC-O(35:4) | 1.23 | 0.676 | 0.24 | 1.56 | 0.4 |
| PC-O(36:1) | 2.58 | 0.608 | 2.35 | 3.5 | 0.9 |
| PC-O(36:2) | 18.6 | 6.6 | 17.9 | 27.8 | 6.8 |
| PC-O(36:3) | 16.65 | 6.15 | 14.5 | 23.8 | 6.1 |
| PC-O(36:4) | 25.25 | 11.1 | 20.3 | 36.1 | 9.2 |
| PC-O(36:5) | 19.95 | 7.525 | 14.2 | 25.8 | 7.3 |
| PC-O(38:1) | 0.462 | 0.366 | 0.003 | 1.09 | 0.2 |
| PC-O(38:2) | 1.11 | 0.449 | 0.885 | 1.6 | 0.4 |
| PC-O(38:3) | <LOD | NA | <LOD | 1.33 | 0.0 |
| PC-O(38:4) | 26.25 | 9.6 | 22 | 40 | 9.6 |
| PC-O(38:5) | 41.45 | 14.725 | 38.5 | 58.6 | 15.1 |
| PC-O(38:6) | 12.5 | 2.775 | 10.8 | 17.5 | 4.5 |
| PC-O(40:2) | 0.953 | 0.544 | 0.51 | 1.44 | 0.3 |
| PC-O(40:3) | 0.735 | NA | <LOD | 0.86 | 0.3 |
| PC-O(40:4) | 3.85 | 1.59 | 3.32 | 6.15 | 1.4 |
| PC-O(40:5) | 7.655 | 2.865 | 6.57 | 10.6 | 2.8 |
| PC-O(40:6) | 6.785 | 2.345 | 5.38 | 10.2 | 2.5 |
| PC-O(40:7) | 4.9 | 0.835 | 3.85 | 5.61 | 1.8 |
| PC-O(40:8) | 0.766 | 0.603 | 0.432 | 1.78 | 0.3 |
| PC-O(42:4) | 0.535 | 0.165 | 0.292 | 0.84 | 0.2 |
| PC-O(42:5) | 2.62 | 1.112 | 1.72 | 3.63 | 1.0 |
| PC-O(42:6) | 1.3 | 0.683 | 0.614 | 1.73 | 0.5 |

IQR, interquartile range; NA, not applicable; LOD, limits of detection.

a limitation of the present study and MS/MS characterization of TG and glycerophospholipids should be considered in the future. However, most likely TG candidates can reasonably be predicted based on known stereochemical structures of these lipids as well as the most common fatty acyl chains present in animals and given that the fatty acid composition of the diet was determined. The impact of diet on the fatty acid composition of plasma TG of several parrot species has been determined [43]. For monoacyl lipids such as non-esterified fatty acids, cholesteryl esters, lyso-PC, the sum composition obviously gives the type of fatty acid. But even then, the position of the double bonds along the fatty acyl chain, the stereochemical positions of the different fatty acyl chains (esterified positions on the glycerol molecule or other head

**Table 14. Plasma monoacylglycerophosphocholines (lysophosphatidylcholines) and monoalkylglycerophosphocholines concentration (μM) in six Quaker parrots (*Myiopsitta monachus*) determined by mass spectrometry.**

| Lipids | Median | IQR | Min | Max | % |
|---|---|---|---|---|---|
| LPC(15:0) | 0.158 | 0.017 | 0.141 | 0.169 | 0.0 |
| LPC(16:0) | 121 | 16.75 | 98.7 | 134 | 19.5 |
| LPC(16:1) | 2.44 | 1.285 | 1.23 | 3.24 | 0.4 |
| LPC(17:0) | 1.004 | 0.072 | 0.825 | 1.08 | 0.2 |
| LPC(17:1) | 0.15 | 0.014 | 0.14 | 0.172 | 0.0 |
| LPC(18:0) | 239.5 | 48 | 177 | 271 | 38.6 |
| LPC(18:1) | 132 | 42.675 | 86.2 | 151 | 21.3 |
| LPC(18:2) | 117 | 17.75 | 86.7 | 136 | 18.8 |
| LPC(20:1) | 1.145 | 0.319 | 0.721 | 1.44 | 0.2 |
| LPC(20:2) | 0.581 | 0.095 | 0.457 | 0.621 | 0.1 |
| LPC(22:5) | 0.686 | 0.106 | 0.608 | 0.849 | 0.1 |
| LPC(22:6) | 4.605 | 1.018 | 3.03 | 5.23 | 0.7 |
| LPC(24:0) | 0.48 | 0.214 | 0.342 | 0.62 | 0.1 |
| LPC-O(16:1) | 0.856 | 0.029 | 0.75 | 0.959 | 0.1 |
| LPC-O(18:0) | 1.59 | 0.39 | 1.42 | 2.11 | 0.3 |
| LPC-O(18:1) | 2.84 | 0.568 | 2.45 | 3.56 | 0.5 |
| LPC-O(18:2) | 1.25 | 0.158 | 1.23 | 1.61 | 0.2 |

IQR, interquartile range.

groups), and the configuration of the carbon-carbon double bonds (cis/trans isomers) are unknown and several lipid species are still represented. In some cases, such as for omega-3 or omega-6 fatty acids, the position of these double bonds is clinically relevant. Untargeted lipidomics is semi-quantitative (peak intensity) and is typically used more for exploratory analysis and discovery work as the analysis is unbiased and not restricted to predetermined panels [44]. Identified lipid species must ideally be confirmed and quantified by targeted methods [44]. For each match with lipid databases, many isobaric species (same or nearly same mass, but different lipids) are also possible, so lipid identification is less certain. Nevertheless, it was used here to complement the targeted panels and give a glimpse of the diversity of other lipids present in Quaker parrot plasma and to highlight which lipid groups and species are important.

**Table 15. Plasma non-esterified sphingoid bases concentration (μM) in six Quaker parrots (*Myiopsitta monachus*) determined by mass spectrometry.**

| Lipids | Median | IQR | Min | Max | % |
|---|---|---|---|---|---|
| Sphinganine (d14:0) | <LOD | NA | <LOD | 0.001 | 0.0 |
| Sphinganine (d16:0) | 0.002 | 0.002 | 0.001 | 0.003 | 0.3 |
| Sphinganine (d17:0) | 0.017 | 0.004 | 0.013 | 0.021 | 2.7 |
| Sphinganine (d18:0) | 0.059 | 0.011 | 0.037 | 0.103 | 9.3 |
| Sphinganine (d18:1) | 0.551 | 0.146 | 0.351 | 0.939 | 87.2 |
| Sphinganine (d18:2) | <LOD | NA | <LOD | <LOD | 0.0 |
| Sphinganine (d20:0) | 0.002 | 0.001 | 0.002 | 0.004 | 0.3 |
| Sphinganine (d20:4) | 0.001 | 0 | 0.001 | 0.001 | 0.2 |

IQR, interquartile range; NA, not applicable; LOD, limits of detection.

**Table 16. Plasma ceramide phosphocholines (sphingomyelins) concentration (µM) in six Quaker parrots (*Myiopsitta monachus*) determined by mass spectrometry.**

| Lipids | Median | IQR | Min | Max | % |
|---|---|---|---|---|---|
| SM(d18:0/12:0) | <LOD | NA | <LOD | <LOD | 0.0 |
| SM(d18:0/16:0) | 3.031 | 0.288 | 2.512 | 3.465 | 1.5 |
| SM(d18:0/24:0) | 20.648 | 1.94 | 15.863 | 28.62 | 10.3 |
| SM(d18:1/06:0) | 0.029 | 0.012 | 0.024 | 0.043 | 0.0 |
| SM(d18:1/12:0) | 0.323 | 0.061 | 0.288 | 0.427 | 0.2 |
| SM(d18:1/14:0) | 1.102 | 0.191 | 0.969 | 1.487 | 0.5 |
| SM(d18:1/16:0) | 72.476 | 8.561 | 60.734 | 81.667 | 36.0 |
| SM(d18:1/17:0) | 0.695 | 0.113 | 0.632 | 0.837 | 0.3 |
| SM(d18:1/18:0) | 28.217 | 3.275 | 20.861 | 30.426 | 14.0 |
| SM(d18:1/18:1) | 0.1 | 0.01 | 0.076 | 0.122 | 0.0 |
| SM(d18:1/18:2) | 0.168 | 0.018 | 0.156 | 0.184 | 0.1 |
| SM(d18:1/20:0) | 17.702 | 1.696 | 16.409 | 21.183 | 8.8 |
| SM(d18:1/20:1) | 35.209 | 4.768 | 31.681 | 44.868 | 17.5 |
| SM(d18:1/20:4) | 0.073 | 0.011 | 0.045 | 0.08 | 0.0 |
| SM(d18:1/22:0) | 20.5 | 2.067 | 15.518 | 22.438 | 10.2 |
| SM(d18:1/22:6) | 1.126 | 0.085 | 1.044 | 1.247 | 0.6 |

IQR, interquartile range; NA, not applicable; LOD, limits of detection.

In addition to the incomplete information on individual lipid species, our investigation of the Quaker parrot lipidome was limited to available targeted panels. In particular, the prenol lipids were not investigated, a group including dolichols, ubiquinones, some fat-soluble vitamins (K and E), and carotenoids. Within investigated lipid categories, some important lipids were also not measured such as most glycerophospholipids other than PC, MGs, and cardiolipins. Several bioactive lipids were reported here, mainly steroid hormones, but the mediator lipidome including fatty acid derivatives was not reported as it was considered outside of the scope of this report, which focused on the high abundance lipid (macrolipidomics). Nevertheless, a complete characterization of the parrot lipidome at a high level of molecular information would require considerable resources and extensive analysis and we believe the snapshot of the lipidome reported here gives a reasonable overview of what could be of relevance for potential clinical research applications. The reported lipidome of other species, mainly mammals, is typically of comparable breadth and details [21, 45, 46]

As the plasma lipidome of other avian species including chickens has not been reported, comparisons can only be made with mammals, in particular humans in which it has been well characterized and quantitative data are available [21, 45–47]. However, methodologies in these other studies were different, thus comparisons were mainly made based on large magnitude of concentration differences and on relative abundance within lipid classes. In addition, the plasma lipidome is heavily influenced by the diet, in particular its fatty acid profile, and the reported lipidome should be interpreted in the context of the diet consumed as humans are omnivorous and Quaker parrots are frugivorous/granivorous. For this reason, the parrot diet was also analyzed. Surprisingly, the Quaker parrot lipidome had many similarities to reported mammalian lipidomes, but also some unique differences. The relative importance of the different lipid categories was similar in parrots and humans on a molar basis (sterols>glycerolipids>glycerophospholipids); however, most lipid group concentrations were much higher in the parrots than in humans [21]. Several very abundant lipids in the human plasma lipidome were also found to be abundant in parrots.

**Table 17. Plasma ceramides concentration (µM) in six Quaker parrots (*Myiopsitta monachus*) determined by mass spectrometry.**

| Lipids | Median | IQR | Min | Max | % |
|---|---|---|---|---|---|
| Cer(d18:1/2:0) | 0.03 | 0.008 | 0.019 | 0.035 | 0.5 |
| Cer(d18:1/4:0) | 0.022 | 0.016 | 0.015 | 0.046 | 0.4 |
| Cer(d18:1/6:0) | 0.005 | 0.001 | 0 | 0.015 | 0.1 |
| Cer(d18:1/8:0) | <LOD | NA | <LOD | <LOD | 0.0 |
| Cer(d18:1/10:0) | 0.003 | 0.002 | 0.002 | 0.007 | 0.1 |
| Cer(d18:1/12:0) | 0.001 | NA | <LOD | 0.003 | 0.0 |
| Cer(d18:1/14:0) | 0.002 | 0 | 0.002 | 0.002 | 0.0 |
| Cer(d18:1/16:0) | 0.064 | 0.151 | 0.007 | 0.245 | 1.1 |
| Cer(d18:1/17:0) | <LOD | NA | <LOD | <LOD | 0.0 |
| Cer(d18:1/18:0) | 0.001 | 0.001 | 0.001 | 0.002 | 0.0 |
| Cer(d18:1/18:1) | <LOD | NA | <LOD | <LOD | 0.0 |
| Cer(d18:1/20:0) | 0.01 | 0.003 | 0.006 | 0.012 | 0.2 |
| Cer(d18:1/20:4) | <LOD | NA | <LOD | <LOD | 0.0 |
| Cer(d18:1/22:0) | 0.783 | 0.16 | 0.675 | 1.387 | 13.1 |
| Cer(d18:1/23:0) | 1.16 | 0.401 | 0.767 | 1.56 | 19.3 |
| Cer(d18:1/24:0) | 1.258 | 0.207 | 0.942 | 1.974 | 21.0 |
| Cer(d18:1/24:1) | 2.609 | 2.237 | 1.078 | 5.3 | 43.5 |
| Cer(d18:1/25:0) | 0.036 | NA | <LOD | 0.368 | 0.6 |
| Cer(d18:1/26:0) | 0.012 | 0.003 | 0.007 | 0.018 | 0.2 |
| Cer(d18:1/28:0) | 0.001 | 0 | 0.001 | 0.001 | 0.0 |

IQR, interquartile range; NA, not applicable; LOD, limits of detection.

The parrot plasma lipidome was dominated by free cholesterol and cholesteryl linoleate [CE(18:2)] on a molar basis, which is also the case in humans. However, CE(18:2) represents only 50% of all CE in humans compared to more than 70% in parrots, but the same number of CE species weren't analyzed. Free cholesterol and all CE concentrations were also much higher

**Table 18. Plasma dihydroceramides concentration (µM) in six Quaker parrots (*Myiopsitta monachus*) determined by mass spectrometry.**

| Lipids | Median | IQR | Min | Max | % |
|---|---|---|---|---|---|
| Dihydroceramide(d18:0/12:0) | <LOD | NA | <LOD | <LOD | 0.0 |
| Dihydroceramide(d18:0/14:0) | <LOD | NA | <LOD | <LOD | 0.0 |
| Dihydroceramide(d18:0/16:0) | 0.001 | 0 | 0.001 | 0.001 | 2.4 |
| Dihydroceramide(d18:0/17:0) | <LOD | NA | <LOD | <LOD | 0.0 |
| Dihydroceramide(d18:0/18:0) | <LOD | NA | <LOD | <LOD | 0.0 |
| Dihydroceramide(d18:0/18:1) | <LOD | NA | <LOD | <LOD | 0.0 |
| Dihydroceramide(d18:0/18:2) | <LOD | NA | <LOD | <LOD | 0.0 |
| Dihydroceramide(d18:0/2:0) | <LOD | NA | <LOD | 0.001 | 0.0 |
| Dihydroceramide(d18:0/20:0) | 0.002 | 0.001 | 0.002 | 0.003 | 4.9 |
| Dihydroceramide(d18:0/20:4) | <LOD | NA | <LOD | <LOD | 0.0 |
| Dihydroceramide(d18:0/22:0) | 0.005 | 0.001 | 0.004 | 0.006 | 12.2 |
| Dihydroceramide(d18:0/22:6) | <LOD | NA | <LOD | 0.001 | 0.0 |
| Dihydroceramide(d18:0/24:0) | 0.033 | 0.014 | 0.02 | 0.042 | 80.5 |

IQR, interquartile range; NA, not applicable; LOD, limits of detection.

**Table 19. Plasma simple glycosphingolipids (cerebrosides and globosides) concentration (μM) in six Quaker parrots (*Myiopsitta monachus*) determined by mass spectrometry.**

| Lipids | Median | IQR | Min | Max | % |
|---|---|---|---|---|---|
| Galactosyl(beta)ceramide(d18:1/12:0) | 0.002 | 0.001 | 0.001 | 0.004 | 0.2 |
| Galactosyl(beta)ceramide(d18:1/16:0) | 0.024 | 0.005 | 0.018 | 0.031 | 1.9 |
| Galactosyl(beta)ceramide(d18:1/22:0) | 0.264 | 0.052 | 0.167 | 0.276 | 20.4 |
| Galactosyl(beta)ceramide(d18:1/24:0) | 0.613 | 0.252 | 0.434 | 0.893 | 47.3 |
| Galactosyl(beta)ceramide(d18:1/24:1) | 0.323 | 0.105 | 0.25 | 0.475 | 24.9 |
| Glucosyl(beta)ceramide (d18:1/18:0) | 0.005 | 0.001 | 0.005 | 0.009 | 0.4 |
| Glucosyl(beta)ceramide(d18:1/18:1) | 0.001 | NA | <LOD | 0.002 | 0.1 |
| Lactosyl(beta)ceramide(d18:1/16:0) | 0.065 | 0.011 | 0.055 | 0.075 | 5.0 |
| Lactosyl(beta)ceramide(d18:1/18:1) | <LOD | NA | <LOD | <LOD | 0.0 |

IQR, interquartile range; NA, not applicable; LOD, limits of detection.

than in humans. However, lathosterol, a marker of cholesterol synthesis, was much lower in Quaker parrots.

For glycerophospholipids, PC(36:2), PC(34:2), and PC(38:4) were some of the most common PCs in both species. PC species identified as abundant on untargeted techniques were also common species in human plasma. Common LPCs in Quaker parrots were also common in humans.

For glycerolipids, a similar TG profile was also observed with TG(52:3) being the most common plasma TG in both species followed by TGs in C52 or C54. The DG profiles were also similar with DG in C36 being the most common.

For sphingolipids, sphingosine was the most common sphingoid base in Quaker parrot as it is the case in humans and mammals in general. SM(d18:1/16:0) was the most common sphingomyelin by far representing about 1/3 of all SM in both species. It is also the most common SM in a number of other mammals [48]. As in humans, the fatty acid distribution in ceramides was quite different from SM with palmitic acid contributing very little compared to very-long

**Table 20. Plasma cholesteryl esters concentration (μM) in six Quaker parrots (*Myiopsitta monachus*) determined by mass spectrometry.**

| Lipids | Median | IQR | Min | Max | % |
|---|---|---|---|---|---|
| CE(16:0) | 479.5 | 113 | 361 | 521 | 3.3 |
| CE(16:1) | 198.5 | 121.55 | 91.1 | 265 | 1.4 |
| CE(17:0) | 11 | 1.595 | 7.07 | 13.1 | 0.1 |
| CE(17:1) | 12.7 | 3.715 | 7.07 | 13.8 | 0.1 |
| CE(17:2) | 6.355 | 0.715 | 4.27 | 8.09 | 0.0 |
| CE(18:1) | 465 | 94 | 243 | 565 | 3.2 |
| CE(18:2) | 10419.5 | 2672.75 | 7704 | 12122 | 72.5 |
| CE(18:3) | 668 | 173.75 | 482 | 752 | 4.6 |
| CE(19:2) | 175.5 | 40.5 | 101 | 226 | 1.2 |
| CE(19:3) | 57.3 | 15.425 | 30.3 | 68.6 | 0.4 |
| CE(20:4) | 960 | 302.75 | 707 | 1377 | 6.7 |
| CE(20:5) | 202 | 64.75 | 120 | 264 | 1.4 |
| CE(22:5) | 151.5 | 80.75 | 116 | 261 | 1.1 |
| CE(22:6) | 562 | 162 | 401 | 674 | 3.9 |

IQR, interquartile range.

**Table 21. Plasma sterols and steroids concentration (μM) in six Quaker parrots (*Myiopsitta monachus*) determined by mass spectrometry.**

| Lipids | Median | IQR | Min | Max |
|---|---|---|---|---|
| 6bOH-cortisol | <LOD | NA | <LOD | <LOD |
| 7-Dehydrocholesterol | 0.042 | 0.007 | 0.034 | 0.068 |
| Aldosterone | 0.001 | NA | <LOD | 0.002 |
| Androstenedione | 0.097 | 0.016 | 0.081 | 0.167 |
| Cholesterol | 9438.4 | 1328.3 | 7715.8 | 12134.2 |
| Cortisol | 0.002 | 0.002 | 0.001 | 0.003 |
| Dehydrodesmosterol | 0.32 | 0.17 | 0.238 | 0.673 |
| Dehydrolathosterol | 3.038 | 0.847 | 2.229 | 3.449 |
| Desmosterol | 0.509 | 0.298 | 0.34 | 1.194 |
| Dihydrolanosterol | 0.054 | 0.01 | 0.046 | 0.078 |
| Lanestenol | <LOD | NA | <LOD | <LOD |
| Lathosterol | 0.268 | 0.058 | 0.2 | 0.512 |
| Zymostenol | <LOD | NA | <LOD | <LOD |
| Zymosterol | 0.109 | 0.039 | 0.077 | 0.24 |

IQR, interquartile range; NA, not applicable; LOD, limits of detection.

chain fatty acids. As in humans and other mammals plasma, ceramides containing C24:1 and C24:0 were the most abundant ceramides in Quaker parrot plasma representing more than 60% of all ceramides [48]. Ceramides are clinically important metabolites and lipid precursors and can also act as bioactive lipids. They have frequently been identified as potential biomarkers for a variety of diseases in humans and mammalian models [20, 49, 50]. Therefore, these similarities in parrots may translate into similar ceramide profiles for lipid-related diseases. Likewise, glycosphingolipids had similar fatty acid compositions than in humans with very-long chain fatty acids predominating.

A difference observed between humans and Quaker parrots pertain to their non-esterified fatty acid profile. In Quaker parrots, palmitic acid was the most common free fatty acid in plasma by far despite these parrots having a dietary fatty acid profile dominated by oleic acid. Free palmitic acid is a potent mediator of lipotoxicity and high levels in Quaker parrots are noteworthy considering their high susceptibility to hepatic lipidosis [2, 51]. By comparison, oleic acid is the main free fatty acid in human plasma. In addition, all concentrations of free fatty acids were far higher to reported concentrations in humans. However, palmitic, stearic, and oleic acids still represented the vast majority of free fatty acids in both species and arachidonic acid and linoleic acids were also the most common PUFAs in both species.

Another marked difference was in the plasma bile acid profile. In parrots, the two major bile acids representing close to 90% of all bile acids were the taurine-conjugated bile acids taurochenodeoxycholic acid and taurocholic acid. The same bile acids also predominate in chickens and turkeys [52]. In humans, these 2 bile acids are also common in plasma but glycine-conjugated bile acids are equally or more common [53, 54] whereas they were at very low to undetectable concentrations in Quaker parrot plasma.

Different lipidomic profiles were detected in female and male Quaker parrots in this study. Although our analysis lacked statistical power to find significant differences between sexes among the individual lipids species, most lipids still appear to be more abundant in females, especially glycerophospholipids and some acyl-carnitines, TGs, and CEs whereas a number of sphingolipids were more abundant in males. These differences are likely explained by the specific female lipid metabolism associated with egg laying (vitellogenesis). Vitellogenesis is

**Table 22. Plasma bile acids concentration (μM) in six Quaker parrots (*Myiopsitta monachus*) determined by mass spectrometry.**

| Bile acid | Median | IQR | Min | Max | % |
|---|---|---|---|---|---|
| 12-Ketochenodeoxycholic acid | <LOD | NA | <LOD | 0.001 | 0.0 |
| 12-Ketolithocholic acid | <LOD | NA | <LOD | 0.001 | 0.0 |
| 3-Oxocholic acid | <LOD | NA | <LOD | <LOD | 0.0 |
| 3b-OH-5-cholestenoic acid | 0.106 | 0.064 | 0.073 | 0.245 | 0.3 |
| 3b7a-diOH-5-cholestenoic acid | 0.028 | 0.006 | 0.024 | 0.036 | 0.1 |
| 6,7-Diketolithocholic acid | 0.001 | NA | <LOD | 0.001 | 0.0 |
| 7-Ketodeoxycholic acid | <LOD | NA | <LOD | <LOD | 0.0 |
| 7-Ketolithocholic acid | <LOD | NA | <LOD | <LOD | 0.0 |
| 7aOH-3-oxo-4-cholestenoic acid | 0.128 | 0.009 | 0.106 | 0.134 | 0.4 |
| α-Muricholic acid | <LOD | NA | <LOD | <LOD | 0.0 |
| Allocholic acid | 0.001 | NA | <LOD | 0.002 | 0.0 |
| Allocholic acid-3-Sulfate | <LOD | NA | <LOD | <LOD | 0.0 |
| Alloisolithocholic acid | 0.001 | NA | <LOD | 0.001 | 0.0 |
| Apocholic acid | <LOD | NA | <LOD | <LOD | 0.0 |
| β-Muricholic acid | <LOD | NA | <LOD | <LOD | 0.0 |
| Chenodeoxycholic acid | 0.008 | 0.001 | 0.007 | 0.009 | 0.0 |
| chenodeoxycholic acid-24-glucuronide | <LOD | NA | <LOD | <LOD | 0.0 |
| chenodeoxycholic acid-3-glucuronide | <LOD | NA | <LOD | <LOD | 0.0 |
| chenodeoxycholic acid-3-Sulfate | <LOD | NA | <LOD | <LOD | 0.0 |
| Cholic acid | <LOD | NA | <LOD | 0.001 | 0.0 |
| cholic acid-3-Sulfate | <LOD | NA | <LOD | <LOD | 0.0 |
| Dehydrocholic acid | <LOD | <LOD | <LOD | <LOD | 0.0 |
| Dehydrolithocholic acid | <LOD | NA | <LOD | 0.001 | 0.0 |
| Deoxycholic acid | 0.001 | NA | <LOD | 0.008 | 0.0 |
| deoxycholic acid-24-glucuronide | <LOD | NA | <LOD | <LOD | 0.0 |
| deoxycholic acid-3-glucuronide | <LOD | NA | <LOD | <LOD | 0.0 |
| deoxycholic acid-3-Sulfate | <LOD | NA | <LOD | <LOD | 0.0 |
| Dioxolithocholic acid | <LOD | NA | <LOD | <LOD | 0.0 |
| Glycoallocholic acid-3-sulfate | <LOD | NA | <LOD | <LOD | 0.0 |
| Glycochenodeoxycholic acid | 0.001 | 0 | 0.001 | 0.002 | 0.0 |
| Glycochenodeoxycholic acid-3Sulfate | <LOD | NA | <LOD | <LOD | 0.0 |
| Glycocholic acid | <LOD | NA | <LOD | 0.001 | 0.0 |
| Glycocholic acid-3-sulfate | <LOD | NA | <LOD | <LOD | 0.0 |
| Glycodehydrocholic acid | <LOD | NA | <LOD | <LOD | 0.0 |
| Glycodeoxycholic acid | <LOD | NA | <LOD | 0.001 | 0.0 |
| Glycodeoxycholic acid-3Sulfate | <LOD | NA | <LOD | <LOD | 0.0 |
| Glycohyocholic acid | <LOD | NA | <LOD | <LOD | 0.0 |
| Glycohyodeoxycholic acid | <LOD | NA | <LOD | <LOD | 0.0 |
| Glycolithocholic acid | <LOD | NA | <LOD | 0.001 | 0.0 |
| GlycolithoCholic acid-3Sulfate | <LOD | NA | <LOD | <LOD | 0.0 |
| Glycoursodeoxycholic acid | <LOD | NA | <LOD | 0.001 | 0.0 |
| Glycoursodeoxycholic acid-3-Sulfate | <LOD | NA | <LOD | <LOD | 0.0 |
| Hyodeoxycholic acid | <LOD | NA | <LOD | <LOD | 0.0 |
| Isodeoxycholic acid | <LOD | NA | <LOD | <LOD | 0.0 |
| Isolithocholic acid | 0.001 | NA | <LOD | 0.001 | 0.0 |
| Lithocholic acid | 0.001 | NA | <LOD | 0.001 | 0.0 |
| lithoCholic acid-24-glucuronide | <LOD | NA | <LOD | 0.001 | 0.0 |

(*Continued*)

**Table 22.** (Continued)

| Bile acid | Median | IQR | Min | Max | % |
|---|---|---|---|---|---|
| lithoCholic acid-3-glucuronide | <LOD | NA | <LOD | <LOD | 0.0 |
| lithoCholic acid-3-Sulfate | <LOD | NA | <LOD | <LOD | 0.0 |
| Murocholic acid | <LOD | NA | <LOD | <LOD | 0.0 |
| γ-muricholic acid | <LOD | NA | <LOD | <LOD | 0.0 |
| Norcholic acid | <LOD | NA | <LOD | <LOD | 0.0 |
| Nordeoxycholic acid | <LOD | NA | <LOD | <LOD | 0.0 |
| Norursodeoxycholic acid | <LOD | NA | <LOD | <LOD | 0.0 |
| Tauro-a-muricholic acid | 0.041 | 0.018 | 0.028 | 0.099 | 0.1 |
| Tauro-b-muricholic acid | <LOD | NA | <LOD | <LOD | 0.0 |
| Tauro-w-muricholic acid | 1.286 | 0.356 | 0.985 | 3.764 | 4.2 |
| Tauroallocholic acid | 2.046 | 0.55 | 1.214 | 4.629 | 6.7 |
| Taurochenodeoxycholic acid | 21.195 | 12.768 | 14 | 46.36 | 68.9 |
| Taurochenodeoxycholic acid-3-sulfate | 0.01 | 0.006 | 0.004 | 0.016 | 0.0 |
| Taurocholic acid | 5.675 | 2.828 | 3.525 | 11.86 | 18.5 |
| Taurodehydrocholic acid | <LOD | NA | <LOD | <LOD | 0.0 |
| Taurodeoxycholic acid | 0.003 | 0.002 | 0.002 | 0.005 | 0.0 |
| Taurodeoxycholic acid-3-sulfate | <LOD | NA | <LOD | <LOD | 0.0 |
| Taurohyocholic acid | 0.002 | 0.001 | 0.001 | 0.003 | 0.0 |
| Taurolithocholic acid | 0.141 | 0.064 | 0.093 | 0.186 | 0.5 |
| TaurolithoCholic acid-3-sulfate | 0.073 | 0.05 | 0.032 | 0.188 | 0.2 |
| Tauroursodeoxycholic acid-3-sulfate | <LOD | NA | <LOD | <LOD | 0.0 |
| Tauroursodexycholic/Taurohyodeoxycholic acid | 0.02 | 0.008 | 0.014 | 0.034 | 0.0 |
| Ursocholic acid | <LOD | NA | <LOD | <LOD | 0.0 |
| Ursodeoxycholic acid | <LOD | NA | <LOD | 0.001 | 0.0 |
| ursodeoxycholic acid-24-glucuronide | <LOD | NA | <LOD | <LOD | 0.0 |
| ursodeoxycholic acid-3-glucuronide | <LOD | NA | <LOD | <LOD | 0.0 |
| ursodeoxycholic acid-3-Sulfate | <LOD | NA | <LOD | <LOD | 0.0 |
| w-muricholic acid | <LOD | NA | <LOD | <LOD | 0.0 |

IQR, interquartile range; NA, not applicable; LOD, limits of detection.

typically associated with increased plasma TG and Glycerophospholipids [55]. The lipids found in eggs are mainly composed of TGs and glycerophospholipids with a lower proportion of cholesterol and cholesteryl esters as shown by published lipidomic analysis of chicken egg yolk [56]. Males also had more Cer(d18:1/18:0), which was identified on both PLS-DA and heatmaps and also more of other long-chain fatty acid sphingolipid species than females. In humans, a number of plasmatic sphingolipids are increased in association with steatohepatitis and fatty liver disease [20]. It is interesting as male Quaker parrots have more than 3 times the prevalence of hepatic lipidosis than female Quaker parrots so this association could also prove to be important in this species [2]. The Quaker parrots in this study were young but either close to sexual maturity or had just reached sexual maturity at time of sampling as some females laid eggs just a few months after the study (sexual maturity is reported to be at 1–2 years of age in this species).

Only 12 Quaker parrots were used in this study with 6 replicates per panel. While this seems like a small sample size, 5–6 animals are typically used for discovery studies on previously uncharacterized plasma lipidomes due to the number of panels and the associated high

**Table 23. Plasma concentration (μM) of non-lipid or lipid-like metabolites and organic acids important in fatty acid metabolic pathways in six Quaker parrots (*Myiopsitta monachus*) determined by mass spectrometry.**

| Analytes | Median | IQR | Min | Max |
| --- | --- | --- | --- | --- |
| γ-Butyrobetaine | 0.274 | 0.073 | 0.205 | 0.423 |
| Free carnitine | 3.786 | 0.398 | 3.223 | 4.605 |
| α-Ketoglutaric acid | 121.75 | 32.02 | 88.23 | 145.50 |
| Citric acid | 281.9 | 43.85 | 213.2 | 341.9 |
| Fumaric acid | 23.82 | 7.9 | 16.63 | 31.17 |
| Glycolic acid | 6.69 | 1.84 | 4.96 | 9.01 |
| Isocitric acid | 8.55 | 1.82 | 5.94 | 12.86 |
| Lactic acid | 11190 | 4860 | 9119 | 18730 |
| Malic acid | 101.39 | 29.86 | 63.60 | 137.70 |
| Pyruvic acid | 620.9 | 162.27 | 415.7 | 786.9 |
| Succinic acid | 57.05 | 23.23 | 25.86 | 89.99 |

IQR, interquartile range.

cost of lipidomic analysis [45–47, 57]. Some of the reported animal plasma lipidomes were also only determined using untargeted techniques. The parrots in this study were relatively young and from a homogeneous colony, therefore the reported lipidome may not capture the variability that is present in different demographic groups and in older parrots.

Another limitation of our study is the accuracy of reported lipid concentrations. While targeted lipidomics is quantitative, it still suffers from accuracy and precision issues. This is due in part to the lack of suitable internal standards for every lipid species, differing ion suppression, collision energy, and ionization efficiency between lipid and fatty acid types, and the lack of standardization of lipidomics techniques [41, 44, 58–60]. Therefore, reported concentrations should be considered as semiquantitative estimates of the true concentrations and comparisons between lipids should only be made within the same lipid classes and using the reported relative values (% in tables or fold changes). Studies have shown up to 30–80% variability/imprecision with various methodology settings and depending on lipid classes [35, 58,

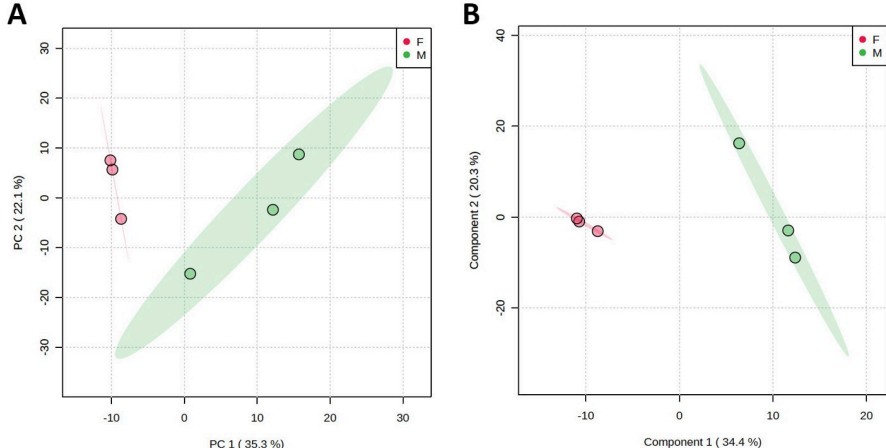

**Fig 2. PCA scores plot (A) and PLS-DA scores plot (B) between the first two components showing clustering of male and female Quaker parrots (*Myiopsitta monachus*) using targeted lipidomics panel between sexes.** Grouping is shown as different colors with their 95% confidence ellipses. The explained variances are shown in brackets.

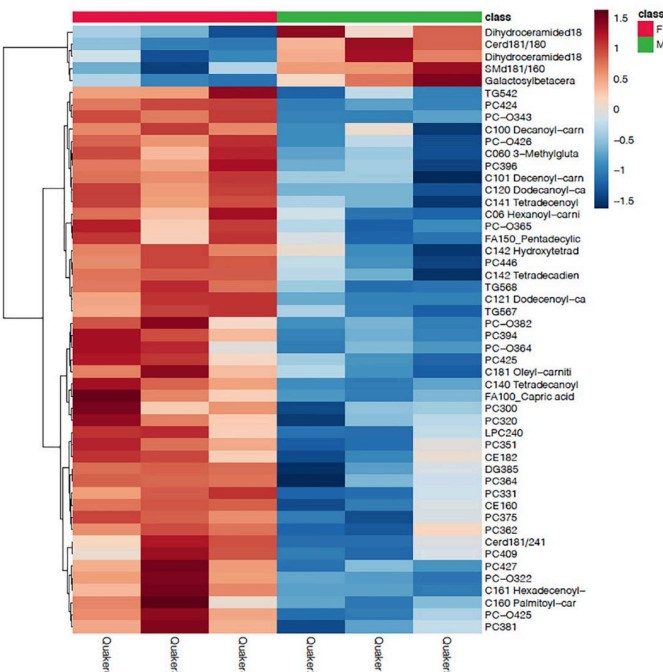

**Fig 3. Heatmap showing clustering of lipid species from targeted lipidomics panels between sexes in Quaker parrots (*Myiopsitta monachus*).** A clustering dendrogram is also present on the left, the different parrots are on the x-axis and the lipid analytes on the y-axis. Only the 50 most important lipids based on their t-test p-values are displayed. It should be noted that most of these lipids did not show significant differences between sexes on univariate analysis. Color coding represents fold changes on normalized plasma concentrations.

60]. The accuracy of the Biocrates Absolute IDQ p400 HR kit used in our study has specifically been investigated across laboratories and showed some inaccuracies with biases of about 50% for most analytes [35]. In this kit, TGs and CEs had the largest biases up to 100–200% in overestimation [35]. This was definitely observed in this study as total cholesterol compounds (free cholesterol + CEs) and total glycerides (TGs + DGs) were largely overestimated compared to known plasma values in this colony of Quaker parrots as measured using conventional techniques [12]. TGs and CEs are neutral lipids, which present specific challenges for absolute quantification using MS-based lipidomics [61].

In conclusion, this study lays the necessary groundwork for further research on the pathophysiology, biomarkers discovery, and pharmacologic intervention of lipid-related diseases in parrots such as atherosclerosis and hepatic lipidosis, which are exceedingly common in captivity. Lipid biomarkers of lipid-related diseases in humans are often lipid mediators such as those derived from polyunsaturated fatty acids (arachidonic acid, ALA, DHA, EPA) as part of the microlipidome or glycerophospholipids such as PCs, PEs, and PIs or metabolite intermediates and precursors of structural lipids such as non-esterified fatty acids, ceramides, LPCs, LPC-Os, DGs, and a variety of small lipid molecules [20, 22, 41, 49, 62, 63]. In Quaker parrots, some of the relative abundance and importance of these lipids is very similar to humans such as for most PCs and DGs while it is very different for others such as non-esterified fatty acids and ceramides. Future research exploring these lipid biomarkers is therefore likely to find both conserved biomarkers across species and species-specific biomarkers. In addition, specific lipidomic signatures of the various psittacine lipid disorders could be investigated as well as lipidomic profiling of parrot plasma with various nutritional and pharmacological interventions

for lipid disorders. Finally, lipidomic fingerprinting between various species of parrots is another area that may help to elucidate species predisposition to certain lipid disorders.

## Acknowledgments

We would like to thank Dr. Omar Zaheer for behavioral training and enrichment of the Quaker parrot colony. We would also like to thank Hagen Inc. for supporting the Quaker parrot colony. Finally, we acknowledge the help of The Metabolomics Innovation Centre for lipidomics analysis.

## Author Contributions

**Conceptualization:** Hugues Beaufrère, Ken D. Stark.

**Data curation:** Hugues Beaufrère.

**Formal analysis:** Hugues Beaufrère.

**Funding acquisition:** Hugues Beaufrère, Sara M. Gardhouse, R. Darren Wood, Ken D. Stark.

**Investigation:** Hugues Beaufrère, Sara M. Gardhouse.

**Methodology:** Hugues Beaufrère, Ken D. Stark.

**Project administration:** Hugues Beaufrère.

**Resources:** Hugues Beaufrère, Ken D. Stark.

**Supervision:** Hugues Beaufrère, R. Darren Wood, Ken D. Stark.

**Visualization:** Hugues Beaufrère.

**Writing – original draft:** Hugues Beaufrère.

**Writing – review & editing:** Hugues Beaufrère, Sara M. Gardhouse, R. Darren Wood, Ken D. Stark.

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
