## [Decision Letter · Decision Letter 0]

10 Aug 2020

PONE-D-20-16607

The plasma lipidome of the Quaker parrot (Myiopsitta monachus)

PLOS ONE

Dear Dr. Beaufrere,

Thank you for submitting your manuscript to PLOS ONE. After careful consideration, we feel that it has merit but does not fully meet PLOS ONE’s publication criteria as it currently stands. Therefore, we invite you to submit a revised version of the manuscript that addresses the points raised during the review process.

I think there is a fair bit of work to be done here but you should be able to address many of the reviewer's comments in a revised version of the manuscript. I am not too concerned about Reviewer 1's comment 5 regarding the sum of the fatty acid composition as I realize this is probably the best resolution you could get. 

We look forward to receiving your revised manuscript.

Kind regards,

Robin D Clugston, Ph.D.

Academic Editor

PLOS ONE

Journal Requirements:

2. In your Methods section, please provide additional details regarding the birds used in your study and ensure you have described the source. For more information regarding PLOS' policy on materials sharing and reporting, see https://journals.plos.org/plosone/s/materials-and-software-sharing#loc-sharing-materials.

3. We note you have included a table to which you do not refer in the text of your manuscript. Please ensure that you refer to Table 3 in your text; if accepted, production will need this reference to link the reader to the Table.

Reviewers' comments:

Reviewer's Responses to Questions

**Comments to the Author**

1. Is the manuscript technically sound, and do the data support the conclusions?

Reviewer #1: No

Reviewer #2: Yes

2. Has the statistical analysis been performed appropriately and rigorously? 

Reviewer #1: No

Reviewer #2: Yes

3. Have the authors made all data underlying the findings in their manuscript fully available?

Reviewer #1: Yes

Reviewer #2: Yes

4. Is the manuscript presented in an intelligible fashion and written in standard English?

Reviewer #1: Yes

Reviewer #2: Yes

5. Review Comments to the Author

Reviewer #1: This is an observational study using target and untargeted lipidome to characterize the lipidome of young male and female healthy Quaker parrots considering them as representative of psittacine birds. Due to advantages of lipidome characterization as base for further study of dyslipidemia and other metabolic dysfunction on these birds the study is an important effort to advance the scientific field.

The study presented a small population sample of 6 males and 6 females’ birds that were used for the lipidome profile. A set of 6 samples (3 male and 3 female) were sent to one center for performance of target lipidome and another set of 6 birds to another center for performance of the untargeted lipidome, as well as for targeted panels for bile acids, sterols, non-esterified fatty acids, acyl carnitines, and sphingolipids.

The authors recognized that although their lipidomic methods could detect free fatty acids and lipids containing only one fatty acid as lysolipids and acylcarnitines and cholesteryl esters, there is a critical limitation of identification of the fatty acid composition of lipids containing two or more fatty acids.

The use of target and untargeted lipidome methods put together to characterize the Quaker parrots lipidome present some issues that need to be clarified, as well as the use of such small sample. The acquisition of only 1 milliliter of blood probably is also a limitation of the study, since the 500 µL appears not enough to divide for shipping for two different samples, and rather the authors divided the sample population (from 12 parrots, 6 were sent to one center and 6 to other).

Major issues:

1- On line 354 the authors state that Identified abundant species in untargeted lipidome did not necessarily correspond to lipid found to be abundant on targeted panels. The untargeted lipidome is presented only on Table 4 and does not appear to add much information to the study, since the other tables on the manuscript appears to come from the target analysis. The authors discussed this issue, but it is not clear the benefit of it for the characterization of the lipidome.

2- The presentation of specific lipids species that were under the limit of detection (LOD) appears to add a lack of accuracy to the lipid characterization proposed. In a few cases as in Table 17 the specie Cer(d18:1/18:1) and Cer(d18:1/20:4) values are presented as LOD although their percentage are 13.1 and 21.0 respectively. An explanation of the importance to show these lipids species should be stated in the discussion.

3- The comparison of gender presents some concerns on the aspect of the power of the analysis. The authors mentioned that only 2 lipids reached the threshold to be considered statistically significant, although the criteria are not specified. This lack of power is probably to the fact of only 3 males and 3 females were compared. However, the multivariate analysis present clear separation of the gender. This creates a problem, since if a sex bias exists in the lipidome, the abundance and relative concentration of the lipidome presented on the several tables of the study must be very influenced by the gender bias and should be explored. Again, the small sample size becomes a problem for a univariate analysis.

4- The hierarchical cluster analysis also shows a clear difference between sexes. It must be stated when presenting this analysis that although a t-test was used to select the 50 more important features that classify the genders, none of them reach statistical significance. To present the result as 50 significant lipids is confusing at the minimum.

5- The use of the sum of the fatty acid composition to represent the lipids is a limitation of the methodology that does not give enough information to the purpose of characterization of the metabolic profile of the Quaker parrots.

Minor issues:

1) Although the authors address some of the major issues on discussion section, all justification related to limitation of the study should be clustered on a Limitation session.

Overall assessment: The issues with the study creates enough problems that are out of the objective of the authors to find metabolic profile of the lipidome of the Quaker parrots, and therefore, the manuscript does not add important information for the advance of the field.

Reviewer #2: The article by Beaufrere, et. al., presents a descriptive and semi-quantitative lipidomic profile of captive Quaker parrot plasma with the goal of establishing the baseline data for future studies. The analytical methods used are highly robust and the authors clearly identified limitations of the study. Because of the descriptive nature of the study, it is understandable that the authors have little scope to draw conclusions with respect to dyslipidemia and lipid accumulation disorders in these birds at this time. More experimental details are needed in the absence of published references. While the methods used were comprehensive and the data presented is extensive, several minor corrections would help improve the manuscript.

1. Some of the methods need to include more details or appropriate references. For example, what are the deuterated standards of bile acids used, what are the chromatographic conditions, and how well are the closely related compounds resolved on the column as well as the mass spectrometric conditions for their detection? This comment applies to all method descriptions where a published procedure is not cited.

2. Description of the preparation of standards to generate standard curves, details about the use of linear regression, etc., can be consolidated in the general methods rather than repeating the same text for each analytical method.

3. Description and definition of lipidomics presented in the third paragraph of the Introduction could be minimized by citing appropriate references. While relatively new, lipidomics is quite an established field to warrant such detailed introduction.

4. Data table descriptions consistently reminded the reader about concentrations as ‘µmol/L’. While this is not inaccurate, the more appropriate designation would be ‘µM’!

5. What is the ‘3-NOH’ in line 218?

6. PLOS authors have the option to publish the peer review history of their article (what does this mean?). If published, this will include your full peer review and any attached files.

Reviewer #1: No

Reviewer #2: **Yes: **Krishna Rao Maddipati

---

## [Author Response · Author response to Decision Letter 0]

3 Sep 2020

Dear Editors and Reviewers,

Thank you for taking the time to review our manuscript and providing insightful comments to improve it. We answered all reviewers’ comments on a point-per-point basis and modified the manuscript accordingly. Please see below our answers to each of the Editor and Reviewers’ comment.

Editor

I think there is a fair bit of work to be done here but you should be able to address many of the reviewer's comments in a revised version of the manuscript. I am not too concerned about Reviewer 1's comment 5 regarding the sum of the fatty acid composition as I realize this is probably the best resolution you could get. 

Authors’ response: this was done

2. In your Methods section, please provide additional details regarding the birds used in your study and ensure you have described the source. For more information regarding PLOS' policy on materials sharing and reporting, see https://journals.plos.org/plosone/s/materials-and-software-sharing#loc-sharing-materials.

Authors’ response: we provided more details on the origin of this Quaker parrot colony.

3. We note you have included a table to which you do not refer in the text of your manuscript. Please ensure that you refer to Table 3 in your text; if accepted, production will need this reference to link the reader to the Table.

Authors’ response: we actually referred to table 3 on the paragraph immediately above at the same time as we referred to figure 1.

Reviewer 1

This is an observational study using target and untargeted lipidome to characterize the lipidome of young male and female healthy Quaker parrots considering them as representative of psittacine birds. Due to advantages of lipidome characterization as base for further study of dyslipidemia and other metabolic dysfunction on these birds the study is an important effort to advance the scientific field.

The study presented a small population sample of 6 males and 6 females’ birds that were used for the lipidome profile. A set of 6 samples (3 male and 3 female) were sent to one center for performance of target lipidome and another set of 6 birds to another center for performance of the untargeted lipidome, as well as for targeted panels for bile acids, sterols, non-esterified fatty acids, acyl carnitines, and sphingolipids.

The authors recognized that although their lipidomic methods could detect free fatty acids and lipids containing only one fatty acid as lysolipids and acylcarnitines and cholesteryl esters, there is a critical limitation of identification of the fatty acid composition of lipids containing two or more fatty acids.

The use of target and untargeted lipidome methods put together to characterize the Quaker parrots lipidome present some issues that need to be clarified, as well as the use of such small sample. The acquisition of only 1 milliliter of blood probably is also a limitation of the study, since the 500 µL appears not enough to divide for shipping for two different samples, and rather the authors divided the sample population (from 12 parrots, 6 were sent to one center and 6 to other).

Authors’ response: These Quaker parrots only weight about 100g so the maximum volume of blood that could be harvested was 1 mL, resulting in 500 uL of plasma maximum. For this reason, we used more birds rather than repeat sampling on the same birds. These birds are also used for other research projects. While it is never a good scientific explanation, the analysis done was also restricted by the amount of funding available for this research as well as the number of birds present in our colony. These multiple lipidomics panels are onerous and we could not practically get a higher sample size. Likely for similar reasons, previous studies have used a similar sample size to determine the plasma lipidome of a species, as discussed in the discussion section. Please see below our responses to other comments.

Major issues:

1- On line 354 the authors state that Identified abundant species in untargeted lipidome did not necessarily correspond to lipid found to be abundant on targeted panels. The untargeted lipidome is presented only on Table 4 and does not appear to add much information to the study, since the other tables on the manuscript appears to come from the target analysis. The authors discussed this issue, but it is not clear the benefit of it for the characterization of the lipidome.

Authors’ response: we run the untargeted lipidomics mainly to have a snapshot of the overall lipid diversity in the plasma of the Quaker parrots in an unbiased way, but also to complement the targeted panels that were missing important categories. An example is the glycerophosphoethanolamines that are both abundant and extremely diverse in the plasma of animals, but were not part of our targeted panels. We also think there is value in reporting the raw data to allow other researchers to explore other metabolites in these birds or use different methods of lipid identification. There are no data on plasma metabolomics/lipidomics in any bird species that is freely available as far as we could find so we would like to report all the data we could gather.

2- The presentation of specific lipids species that were under the limit of detection (LOD) appears to add a lack of accuracy to the lipid characterization proposed. In a few cases as in Table 17 the specie Cer(d18:1/18:1) and Cer(d18:1/20:4) values are presented as LOD although their percentage are 13.1 and 21.0 respectively. An explanation of the importance to show these lipids species should be stated in the discussion.

Authors’ response: we are very sorry, but we made a huge mistake on all the percentages on that table (the ceramide table), which did not match the corresponding lipid. We corrected this table and this changed our comparison with the human lipidome in the discussion (which was based on those percentages), which we also corrected. Thank you for catching this mistake.

We hope that with that corrected table, the LOD makes more sense. As it represents the lower limit of detection of the assays, it simply means that these lipid species may still be present, but not detected with the methods used in our research. Practically, this can be interpreted as those lipid species being minor lipid species within the respective lipid categories. We believe that it should still be reported that we tried to measure these lipids and they were at very low concentrations in the plasma.

We checked all other tables for similar mistakes and could not find any.

3- The comparison of gender presents some concerns on the aspect of the power of the analysis. The authors mentioned that only 2 lipids reached the threshold to be considered statistically significant, although the criteria are not specified. This lack of power is probably to the fact of only 3 males and 3 females were compared. However, the multivariate analysis present clear separation of the gender. This creates a problem, since if a sex bias exists in the lipidome, the abundance and relative concentration of the lipidome presented on the several tables of the study must be very influenced by the gender bias and should be explored. Again, the small sample size becomes a problem for a univariate analysis.

Authors’ response: as mentioned by the reviewer, there is definitely a lack of power in this study for such a comparison when assessing hundreds of lipids on only 6-12 parrots. The criteria for significance were clearly mentioned in the MM (data normalization, false discovery rate of 5%) and the q-values of these differences are reported. Because of the lack of power, we could not identify more individual lipids that were different between sexes, we made this clearer in the discussion. The multivariate analysis is more useful in our opinion here as it allows to better summarize our high-dimensional data. The PCA seemed to explain most of the variance and therefore seems to be a valid approach.

We obviously did not have the sample size to stratify these reported concentrations and if such stratification is warranted, it would not affect all analytes anyway. Our goal was not to report reference intervals for all these lipids, but just some concentrations to get a sense of what is abundant and what is not. The sample size did not allow further exploration of sex-based differences in the lipidome so this part of the results should be considered as preliminary information.

4- The hierarchical cluster analysis also shows a clear difference between sexes. It must be stated when presenting this analysis that although a t-test was used to select the 50 more important features that classify the genders, none of them reach statistical significance. To present the result as 50 significant lipids is confusing at the minimum.

Authors’ response: You are right, we clarified that in the figure caption. We remove the term “significant”.

5- The use of the sum of the fatty acid composition to represent the lipids is a limitation of the methodology that does not give enough information to the purpose of characterization of the metabolic profile of the Quaker parrots.

Authors’ response: It is common for the description of the lipidome of a species to report sum composition for complex lipids as panels that go beyond are typically not commercially available for targeted analysis. We added a statement that this is a limitation and should be addressed in the future. We have also added a reference to the literature indicating the relationship between diet and fatty acid composition in parrot plasma lipids and that the diet used in this study could help interpret the sum compositional data.

Minor issues:

1) Although the authors address some of the major issues on discussion section, all justification related to limitation of the study should be clustered on a Limitation session.

Authors’ response: we would argue that the current organization of our discussion flows better as limitations are highlighted when associated with a particular concept. Also most of the general limitations were discussed together in a dedicated section at the end of the discussion. 

Overall assessment: The issues with the study creates enough problems that are out of the objective of the authors to find metabolic profile of the lipidome of the Quaker parrots, and therefore, the manuscript does not add important information for the advance of the field.

Authors’ response: our objectives was to give an overall overview of the plasma lipidome of the Quaker parrot with the presentation of lipid concentrations as well. As mentioned in the discussion, the similar articles on the lipidome of mammals typically use a similar sample size and a lower amount of information, typically only untargeted lipidomics with no quantitative data. Therefore our manuscript reports a more comprehensive lipidome for a species than most previous articles on mammals and is also the first to report the plasma lipidome of a non-mammalian species.

Our limitations are mainly based on sample sizes and the lack of more complete panels. However, it is difficult to come up with a homogeneous colony of these kinds of animals and due to the expense of lipidomics panels and the requirements of multiple panels to report a lipidome, there is also a financial limitation to what can be achieved to report the plasma lipidome of an avian species.

We would argue that this article adds important information for the advancement of the field of avian medicine and avian clinical pathology, which are the fields we are most interested in.

Reviewer #2: The article by Beaufrere, et. al., presents a descriptive and semi-quantitative lipidomic profile of captive Quaker parrot plasma with the goal of establishing the baseline data for future studies. The analytical methods used are highly robust and the authors clearly identified limitations of the study. Because of the descriptive nature of the study, it is understandable that the authors have little scope to draw conclusions with respect to dyslipidemia and lipid accumulation disorders in these birds at this time. More experimental details are needed in the absence of published references. While the methods used were comprehensive and the data presented is extensive, several minor corrections would help improve the manuscript.

1. Some of the methods need to include more details or appropriate references. For example, what are the deuterated standards of bile acids used, what are the chromatographic conditions, and how well are the closely related compounds resolved on the column as well as the mass spectrometric conditions for their detection? This comment applies to all method descriptions where a published procedure is not cited.

Authors’ response: more information was added to the MM as supplied by The Metabolomics Innovation Centre. Additional references were also added.

2. Description of the preparation of standards to generate standard curves, details about the use of linear regression, etc., can be consolidated in the general methods rather than repeating the same text for each analytical method.

Authors’ response: we know it looks a little cumbersome like this, but the different analyses were done by external laboratories and they each supply their MM details as par of a fee-per-service analysis. We just did not want to modify the details of their protocol not to introduce mistakes or inaccuracies. The MM was further reviewed and completed by the TMIC. As there is no length limit for articles published in PlosOne, this hopefully should not be a big issue as long as the MM is complete.

3. Description and definition of lipidomics presented in the third paragraph of the Introduction could be minimized by citing appropriate references. While relatively new, lipidomics is quite an established field to warrant such detailed introduction.

Authors’ response: we agree with the reviewer on this point. However, our article also has a potential audience with avian veterinarians, small animal veterinarians, veterinary clinical pathologists and veterinary researchers who may not necessarily be very familiar with lipidomics. For this reason and because PlosOne does not limit the size of manuscripts, we would like to keep these introductory statements for this targeted audience.

4. Data table descriptions consistently reminded the reader about concentrations as ‘µmol/L’. While this is not inaccurate, the more appropriate designation would be ‘µM’!

Authors’ response: this was changed.

5. What is the ‘3-NOH’ in line 218?

Authors’ response: this was a typo and should have been 3-NPH.

---

## [Decision Letter · Decision Letter 1]

28 Sep 2020

The plasma lipidome of the Quaker parrot (Myiopsitta monachus)

PONE-D-20-16607R1

Dear Dr. Beaufrere,

We’re pleased to inform you that your manuscript has been judged scientifically suitable for publication and will be formally accepted for publication once it meets all outstanding technical requirements.

Kind regards,

Robin D Clugston, Ph.D.

Academic Editor

PLOS ONE

Additional Editor Comments (optional):

Reviewers' comments:

Reviewer's Responses to Questions

**Comments to the Author**

1. If the authors have adequately addressed your comments raised in a previous round of review and you feel that this manuscript is now acceptable for publication, you may indicate that here to bypass the “Comments to the Author” section, enter your conflict of interest statement in the “Confidential to Editor” section, and submit your "Accept" recommendation.

Reviewer #1: All comments have been addressed

Reviewer #2: All comments have been addressed

2. Is the manuscript technically sound, and do the data support the conclusions?

Reviewer #1: Yes

Reviewer #2: Yes

3. Has the statistical analysis been performed appropriately and rigorously? 

Reviewer #1: Yes

Reviewer #2: Yes

4. Have the authors made all data underlying the findings in their manuscript fully available?

Reviewer #1: Yes

Reviewer #2: Yes

5. Is the manuscript presented in an intelligible fashion and written in standard English?

Reviewer #1: Yes

Reviewer #2: Yes

6. Review Comments to the Author

Reviewer #1: The authors have addressed the major and minor issues found in the previous submission adequately and have edited the tables and point out in the discussion the limitations of the analysis when dealing with a small sample size study. Due to the novelty of the characterization of the Quaker parrots lipidome and the satisfactory answers to my questions I suggest the publication of the manuscript.

Reviewer #2: The authors addressed all my concerns. The study provides basis for additional work in avian biology, especially to improve captive bird management.

7. PLOS authors have the option to publish the peer review history of their article (what does this mean?). If published, this will include your full peer review and any attached files.

Reviewer #1: **Yes: **Daniel Contaifer Junior

Reviewer #2: **Yes: **Krishna Rao Maddipati

---

## [Editor Report · Acceptance letter]

19 Nov 2020

PONE-D-20-16607R1 

The plasma lipidome of the Quaker parrot *(Myiopsitta monachus)*

Dear Dr. Beaufrere:

I'm pleased to inform you that your manuscript has been deemed suitable for publication in PLOS ONE. Congratulations! Your manuscript is now with our production department. 

Kind regards, 

on behalf of

Dr. Robin D Clugston 

Academic Editor

PLOS ONE